# Anomaly Detection Based on Unsupervised Disentangled Representation Learning in Combination with Manifold Learning

## Abstract

Identifying anomalous samples from highly complex and unstructured data is a crucial but challenging task in a variety of intelligent systems. In this paper, we present a novel deep anomaly detection framework named AnoDM (standing for **Ano**maly detection based on unsupervised **D**isentangled representation learning and **M**anifold learning). The disentanglement learning is currently implemented by $\beta$-VAE for automatically discovering interpretable factorized latent representations in a completely unsupervised manner. The manifold learning is realized by t-SNE for projecting the latent representations to a 2D map. We define a new anomaly score function by combining $\beta$-VAE's reconstruction error in the raw feature space and local density estimation in the t-SNE space. AnoDM was evaluated on both image and time-series data and achieved better results than models that use just one of the two measures and other deep learning methods.

## 1 Introduction

Detecting anomalies in data flow of modern intelligent systems is an important but challenging problem. Formally speaking, anomaly detection problems can be statistically viewed as identifying outliers having low probabilities from the modelling of data distribution $p(\boldsymbol{x})$. Practically, since statistical modelling of the data is often difficult, it degenerates to domain description (Tax & Duin, 1999) or supervised prediction (Gornitz et al., 2013) problems in some cases. The exact explanation of an anomalous data point depends on the specific domain of focus. In data centers, it probably indicates an attempt of cyber intrusion. In recognition systems, it could be an adversarial attack. In biomedical information systems, it means possible onset of certain diseases. In Internet of Things (IoT) systems, it may represent a hardware failure or alarming event captured by sensors. An anomalous sample is not always associated with negativity. Sometimes, it leads to novel discoveries in scientific explorations.

However, from the data analytics perspective, anomaly detection is a difficult task due to the following reasons. (1) Many forms of data, e.g., images, text, and other types of sequences, are often highly unstructured and complex. How can these data be well represented and high-level information be extracted by an algorithm? (2) The sample sizes of modern data sets are often extremely large and most of them are unlabelled. Unfortunately, traditional methods do not scale and perform well on these data. (3) When data of multiple modalities are naturally available for same events in a system, a robust and precise algorithm needs to be designed to integrate these information for system diagnosis or decision making. (4) Many intelligent systems, such as IoTs, require real-time detection and reaction of abnormal events to avoid costly and irrevocable damages. Thus, anomaly monitoring algorithms to be designed in these platforms must be highly efficient. In summary, anomaly detection raises challenges in representability, scalability, multimodality, and time complexity.

Deep learning (LeCun et al., 2015) offers great potentials to overcome these challenges. (1) Representation learning mechanisms (such as convolution for images, embedding for discrete symbols, and recurrence for time-series) have been developed in supervised and unsupervised deep models to consider the nature of specific types of input samples and encode them into vectors of continuous values as corresponding latent representations. (2) Most deep learning models are trained using stochastic gradient descent that splits a giant training set into mini-batches. Thus, learning be-

comes unrestricted and blessed by a large sample size. Particularly, stochastic variational inference (Hoffman et al., 2013; Zhang et al., 2018) has successfully enabled scalable learning and inference for deep generative models (DGMs) on a vast amount of unlablled data. (3) The development of deep learning programming packages, such as PyTorch (Paszke et al., 2017) and TensorFlow (Abadi et al., 2016), greatly eases the assembly of multiple network components (corresponding to different modalities) together for multimodal representation learning (Li et al., 2018b). (4) Once a deep model is learned, the inference or encoding step is very efficient, thanks to the highly parallel computing architectures and techniques.

In some applications, if the domain of anomalous and normal samples is well defined, anomaly detection can be reduced to binary classification problems. However, in many situations, either the domain of anomalous samples cannot be fully understood or modelled, or the domain of the normal samples is too complicated to be modelled in one class. DGMs are more suitable than supervised methods in such cases. DGMs are concerned with the joint distribution of visible and latent variables with a hierarchy of stochastic (and deterministic) layers. With proper emphasis on disentanglement of latent representations, DGMs have the potential of dissecting hidden factors that are key to sample generation. Unsupervised disentangled representation learning (Bengio et al., 2013) renders several benefits. (1) It helps better understand our data, providing a path towards explainable AI. (2) It gives a better control on the generation process of novel samples. (3) The disentanglement of latent factors may provide an opportunity to distinguish anomalies based on the landscape of latent space, which is our interest in this paper. It has been shown that the likelihood of a data point $p(\boldsymbol{x})$ estimated in DGM is not a reliable measure for detecting abnormal samples (Nalisnick et al., 2019). Instead, reconstruction error is widely used as an anomaly score function (An & Cho, 2015).

As a variant of variational autoencoder (VAE) (Kingma & Welling, 2014), $\beta$-VAE (Higgins et al., 2017) is designed for unsupervised discovery of interpretable factorized latent representations from raw image data. An adjustable hyperparameter $\beta$ is introduced to balance the extent of learning constraints (a limit on the capacity of the latent information channel and an emphasis on learning statistically independent latent factors) and reconstruction accuracy. It was demonstrated that $\beta$-VAE with appropriately tuned value of $\beta$ (when $\beta > 1$) qualitatively outperforms VAE (when $\beta = 1$, $\beta$-VAE is exactly VAE). Burgess et al. (2018) proposed a modification to the training regime of $\beta$-VAE by progressively increasing the information capacity of the latent code during training. This modification facilitates the robust learning of disentangled representations in $\beta$-VAE, without the previous trade-off in the reconstruction accuracy. Hoffman et al. (2017) introduced a reformulation of $\beta$-VAE for $0 < \beta < 1$. They argued that, within in this range, training $\beta$-VAE is equivalent to optimizing an approximate log-marginal likelihood bound of VAE under an implicit prior.

Manifold learning is a family of nonlinear dimensionality reduction techniques. The t-distributed stochastic neighbor embedding (t-SNE) (van der Maaten & Hinton, 2008) is an unsupervised manifold learning method primarily used for data exploration and visualization by approximating high-dimensional data distribution using a two or three-dimensional map that could preserve local and certain global structures of the data. The use of t-SNE for anomaly detection has been sceptical (van der Maaten & Hinton, 2008). However, no comprehensive investigation has been made in this topic. Taking advantages of both disentangled representation learning (using $\beta$-VAE as an implementation) and low-dimensional manifold learning (using t-SNE as an implementation), we propose a novel anomaly detection approach named **AnoDM**, standing for *Anomaly detection based on unsupervised Disentangled representation learning and Manifold learning*. We introduce a new anomaly score function by combining: (1) $\beta$-VAE's reconstruction error, and (2) distances between latent representations of test points and training points in t-SNE map. AnoDM is a general framework, thus any disentangled representation learning and manifold learning techniques can be applied. The choice of a lower-level encoding scheme in $\beta$-VAE depends on data type of interest. For image data, deterministic convolutional network (CNN) is used in the encoder. In case of time series (sequence) data, we design an improved version of $\beta$-VAE by replacing CNN with temporal convolutional network (TCN) (Bai et al., 2018), a generic architecture for convolutional sequence prediction, in the encoder. We incorporate TCN as part of the encoder, because Bai et al. (2018) have shown that TCN outperforms canonical recurrent networks such as LSTMs (Hochreiter & Schmidhuber, 1997) across a range of supervised learning tasks and recommended that CNN should be regarded as the first method to try for sequence modeling tasks. Regarding the decoding architecture, we simply choose CNN, because by choosing a simpler CNN architecture as a part of the decoder, the model can achieve a comparable even better performance but take much less running time.

The contributions of this paper are summarized as follows. (1) We comprehensively explore the capacity of unsupervised disentangled representation learning, using $\beta$-VAE as an implementation, for anomaly detection. (2) We thoroughly investigate the potential of manifold learning for outlier identification by taking the disentangled latent representations from $\beta$-VAE as input to t-SNE. To the best of our knowledge, this is the first attempt to explore t-SNE for anomaly detection. (3) For sequence anomaly detection, instead of using prevailed recurrent networks (such as LSTM), as a practical contribution, we adopt an improved convolution architecture (TCN) to capture the temporal dependency in the encoder in unsupervised way.

## 2 RELATED WORK

In the big data era, the development of deep learning models, especially DGMs, flourishes due to the need of modelling and analyzing massive amount of unstructured data (such as images, time-series, graphs, text, etc.) generated in many application domains. Designing DGM-based solutions for anomaly detection becomes an important topic. Since DGMs, such as VAE (Kingma & Welling, 2014), and deep belief net (DBN) (Hinton et al., 2006; Li & Zhu, 2018a), aim at modelling the joint distribution of visible and latent variables (that is $p(\boldsymbol{x}, \boldsymbol{h})$), their likelihood $p(\boldsymbol{x})$ by marginalizing out $\boldsymbol{h}$ may serve as an abnormality indicator. However, unlike exponential family restricted Boltzmann machines (exp-RBMs) (Li & Zhu, 2018b), exact likelihood is unavailable for most DGMs. Alternatively, reconstruction error serves as an abnormality measure based on the intuition that out-of-distribution samples can be reconstructed badly (An & Cho, 2015). Some deep hybrid methods (e.g. VAE+OCSVM (Andrews et al., 2016) and DBN+OCSVM (Erfani et al., 2016)), successfully combine classical one-class support vector machine (OCSVM; or kernel-based support vector domain description (SVDD)) with DGMs by using DGMs to learn latent representations of samples and using OCSVM to detect abnormal data points. However, these methods face the challenge of scalability, because the size of kernel matrices in dual form of SVDD is quadratic of sample size.

Generative adversarial net (GAN) has also been applied for anomaly detection (Schlegl et al., 2017). Since there is no encoder in GAN, Deecke et al. (2019) presented the ADGAN algorithm based on the availability of a good representation of a sample in latent space of its generator by assuming that the generator is able to effectively capture the distribution of the training data. Li et al. (2018a) proposed the GAN-AD method for cyber-physical systems (CPSs). It distinguishes fake data from actual data by taking into consideration of both discrimination loss calculated by the trained discriminator and residual loss between reconstructed and actual test data.

Furthermore, DGM-based algorithms are also devised to detect anomaly problem on sequence data (e.g. LSTM-VAE (Park et al., 2017) and GAN-AD (Li et al., 2018a)). Conventionally, canonical recurrent networks (such as LSTM and GRUs (Cho et al., 2014)) are considered as the dedicated methods for sequence modeling. Some recent studies have also claimed that there was no architecture that could consistently beat LSTM in some typical sequence modelling tasks (Jozefowicz et al., 2015; Greff et al., 2015; Melis et al., 2017). On the other hand, some other researchers insist that CNN (LeCun et al., 1989) should be considered as more appropriate choice for sequences. Inspired by more recent CNN-based sequence modelling (such as machine translation (Gehring et al., 2016; 2017) and language modeling (Dauphin et al., 2016)), Bai et al. (2018) conducted a systematic evaluation of generic convolutional and recurrent architectures for sequence modelling across a broad range of tasks that are commonly used to benchmark recurrent networks, and concluded that convolutional networks, rather than recurrent networks, should be respected as a "natural starting point for sequence modelling tasks".

## 3 METHOD

In this paper, we propose a novel generic anomaly detection framework named AnoDM, which the first time combines unsupervised disentangled representation learning (implemented using $\beta$-VAE as an example) and low-dimensional manifold learning (currently using t-SNE as implementation) together to detect outliers via effectively taking the advantages of reconstruction at raw feature space and disentangled latent distribution in t-SNE map. Figure 1 shows the architecture of AnoDM which includes two main phases: (1) unsupervised disentangled representation learning and (2) anomaly detector. After $\beta$-VAE is learned using unlablled training normal samples, it then can be employed

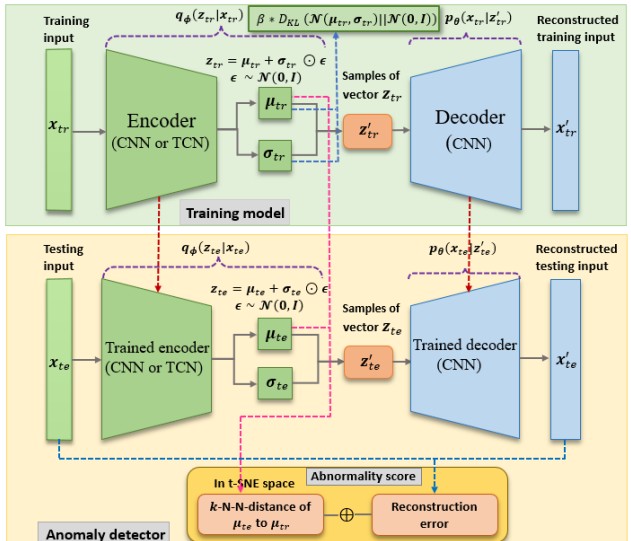

Figure 1: Architecture of AnoDM implemented by $\beta$-VAE and t-SNE for anomaly detection. First, $\beta$-VAE is learned using normal training data (upper part of the framework). Then it is employed by the anomaly detector (lower part of the framework) to efficiently obtain latent encodings of training samples (or a representative subset from the training set) and test samples, and corresponding reconstructed versions using the decoder. Meanwhile, t-SNE is used to map the deterministic latent embeddings ($\boldsymbol{\mu}_{tr}$ for training and $\boldsymbol{\mu}_{te}$ for test samples) to the 2D space (we call it t-SNE space or map), such that the average distance between the 2D representation of a test sample and its $k$ nearest neighbors from the 2D representations of training samples is calculated. Finally, the distance is combined with the reconstruction error of the test sample to define its anomaly score.

by the anomaly detector to efficiently obtain latent encoding and reconstructed version of a sample. Once latent embeddings of both training samples (or a representative subset from the training data) and a test sample are obtained, t-SNE is used to map the latent representations of these samples further to the 2-dimensional space (called t-SNE space or map), such that the average distance between the 2D representation of the test sample and its $k$ nearest neighbors from the 2D representations of training samples is calculated. Finally, this distance is combined with the reconstruction error of the test sample to define its anomaly score. The essential parts of this framework are discussed below in details. Full AnoDM approach is given in Algorithm 1 in Appendix E.

### 3.1 UNSUPERVISED DISENTANGLED REPRESENTATION LEARNING

The unsupervised disentangled representation learning component in our architecture is implemented but not limited by $\beta$-VAE (Higgins et al., 2017; Burgess et al., 2018; Hoffman et al., 2017; Mathieu et al., 2019). A self-inclusive description of $\beta$-VAE is provided in Appendix B. The objective function to be maximized for $\beta$-VAE is defined as (Burgess et al., 2018):

$$\mathcal{L} = \mathbb{E}_{q_\phi(\boldsymbol{z}|\boldsymbol{x})}[\log p_\theta(\boldsymbol{x}|\boldsymbol{z})] - \beta|D_{\mathrm{KL}}(q_\phi(\boldsymbol{z}|\boldsymbol{x})||p(\boldsymbol{z})) - C|. \qquad (1)$$

The first term of this objective function corresponds to reconstruction error in the raw feature space. The KL divergence characterizes the discrepancy between approximate posterior and isotropic prior of latent representations. A small discrepancy between them indicates high disentanglement of latent representations in independent variables. $C$ is a hyperparameter which is used to improve the quality of reconstructed images. The loss function of the original $\beta$-VAE proposed in (Higgins et al., 2017) does not have this hyperparameter. The value of $\beta$ trades off reconstruction error and disentanglement. Unlike (Higgins et al., 2017) and (Burgess et al., 2018), we consider $\beta > 0$ rather than just $\beta > 1$, because it is unnecessary to bound the value of $\beta$ by 1, $\beta > 0$ allows us search for a more appropriate disentanglement. The special case $\beta = 0$ could make the model learning very unstable, because the variance of inference distribution loses control. It worth highlighting that other unsupervised disentanglement models can be used as well in AnoDM. For example, Mathieu et al. (2019) interpreted disentanglement as decomposition instead of independence by adding an additional regularization term to reduce the discrepancy between the aggregate posterior and a desired structured

prior. However, designing a properly structured prior could be practically challenging. The adoption of $\beta$-VAE in our framework is sufficient to prove the concept that unsupervised disentanglement helps anomaly detection.

## 3.2 EFFECTIVITY OF T-SNE ALGORITHM

In addition to $\beta$-VAE's reconstruction error, we use the average distance between a test sample and its $k$-nearest neighbors from the collection of training samples in the t-SNE map to score the outlierness of a test sample. As discussed in Appendix C, t-SNE is significantly influenced by perplexity. The nature of complexity in data distributions makes it impossible to utilize a uniform criteria to define optimal perplexity for all data. Moreover, Wattenberg et al. (2016) mentioned several weaknesses of t-SNE, for examples, (1) it naturally expands dense clusters and contracts sparse ones, evening out cluster sizes, and (2) distances between clusters might not reflect global geometry. However, it is likely that $k$-nearest neighbors still work for local clumps, because, with proper value of perplexity, local topological information of latent distributions can be preserved by the t-SNE plot. Thus, in t-SNE space, the measure of $k$-nearest neighbors is more suitable than full density estimation which is very sensitive with the sizes of clusters. Furthermore, as to be shown in Section 4, distancing in the 2D t-SNE map could be more robust than distancing in $\beta$-VAE's latent space where many non-determinative factors may influence the calculation of distances. Finally, it worth clarifying that we do not directly learn a 2D latent representation from $\beta$-VAE, because it will bottleneck too much the information flow for reconstruction. Instead, a lower-dimensional representation is learned through t-SNE for density estimation only.

## 3.3 DETERMINISTIC OR STOCHASTIC LATENT REPRESENTATIONS FOR T-SNE

Either the mean $\boldsymbol{\mu}$ or a sample $\boldsymbol{z}$ from the approximate inference distribution $q(\boldsymbol{z}|\boldsymbol{x})$ can be passed to t-SNE to calculate the $k$-NN distance of a test sample. There is trivial difference between performances achieved by these two methods in our framework. Generally, the $\boldsymbol{\mu}$-based method achieved slight better performance. The comparison of these two methods can be found in Table 2 in appendices. Furthermore, from the latent representations' t-SNE maps (see Figure 9 in appendices), one can interestingly see that when $\beta$ is small (not overly large), the t-SNE maps for both methods are quite similar. As $\beta$ becomes overly large, some normal classes can still form their own clusters (even though some similar classes, such as classes 3 and 8 in MNIST, tend to mingle together) in $\boldsymbol{\mu}$-based method, but in $\boldsymbol{z}$-based method all classes are entangled with each other. Same phenomena can be observed on the other datasets (see Figures 10 and 11 in appendices). Therefore, the $\boldsymbol{\mu}$-based method is used in current design of AnoDM.

## 3.4 TCN ENCODER FOR UNSUPERVISED SEQUENCE MODELLING

Bai et al. (2018) distilled superior design in convolutional network into a simple architecture and referred it as a temporal convolutional network (TCN) with two distinctive characteristics: (1) the convolutions in the architecture are causal, and (2) the architecture can take a sequence of any length and map it to an output vector of fixed length, just as with an RNN. Bai et al. (2018) also explained that TCNs capture significantly longer history than recurrent networks. Inspired by Bai et al. (2018), we replace CNN with TCN in the encoder of $\beta$-VAE when evaluating the proposed AnoDM framework on time-series data, while we still use CNN in the decoder, because our preliminary experiments demonstrated that keeping decoder as simpler CNN can help achieve comparable even better results, and take much less computing time. The architecture of TCN used in this paper is depicted in Figure 7 of appendices. In this TCN, we set kernel size to 4 and dilation factors to $[1, 2, 4, 8, 16, 32]$. In Section 4, the comparison among TCN, CNN, and LSTM encoders in our framework also shows that TCN outperforms CNN and particularly LSTM to a great extent for ECG signal anomaly detection.

## 3.5 ANOMALY SCORE FUNCTION IN ANODM

In the anomaly detector, the reconstruction error of a test sample in the original feature space and the average distance from its $k$-nearest-neighbors in training samples within the 2D t-SNE map are

combined as a final anomaly score function:

$$\mathcal{S}_{\beta\text{VAE+tSNE}}(\boldsymbol{x}_{\text{te}}) = \alpha\mathcal{D}_{\text{RE}}(\boldsymbol{x}_{\text{te}}) + (1-\alpha)\mathcal{D}_{\text{tSNE}}^k(\boldsymbol{x}_{\text{te}}), \qquad (2)$$

where the first term is defined using normalized squared error (NSE):

$$\mathcal{D}_{RE}(\boldsymbol{x}_{\text{te}}) \triangleq \text{NSE}(\boldsymbol{x}_{\text{te}}, \boldsymbol{x}_{\text{te}}') = \frac{\|\boldsymbol{x}_{\text{te}} - \boldsymbol{x}_{\text{te}}'\|_2^2}{\|\boldsymbol{x}_{\text{te}}\|_2}, \qquad (3)$$

where $\boldsymbol{x}_{\text{te}}$ is a test sample, and $\boldsymbol{x}_{\text{te}}'$ is its reconstructed version by sending the stochastic latent encoding through the decoder of $\beta$-VAE. The second term in Equation (2) is defined using $\beta$-VAE's deterministic latent encoding (mean from the encoder of learned $\beta$-VAE) as input to $t$-SNE:

$$\mathcal{D}_{\text{tSNE}}^k(\boldsymbol{x}_{\text{te}}^{(i)}) \triangleq \frac{1}{k} \sum_{j \in N(i,k)} \|\boldsymbol{l}_{\text{te}}^{(i)} - \boldsymbol{l}_{\text{tr}}^{(j)}\|_2, \qquad (4)$$

where $\boldsymbol{l}_{\text{te}}^{(i)}$ is the 2D representation of the $i$-th test sample in t-SNE map, $N(i,k)$ is the set of indices of $\boldsymbol{l}_{\text{te}}^{(i)}$'s $k$ nearest neighbors from training samples' 2D representations $\boldsymbol{l}_{\text{tr}}$ in t-SNE map. In Equation (2), $\alpha \in [0, 1]$ is the combination hyperparameter such that the two terms can effectively complement each other. To allow the anomaly score function to achieve its full potential, $\alpha$ value should be sensitively searched, because the values of $\mathcal{D}_{\text{RE}}$ and $\mathcal{D}_{\text{tSNE}}^k$ can stay at different magnitudes, a very small change of $\alpha$ value may dramatically alter the contributions of these two terms. Alternatively, the distance score in Equation 4 can be normalized by average distance of training samples in t-SNE map, which may alleviate the magnitude difference, thus ease search of optimal value of $\alpha$. The use of this normalized distance is investigated in Appendix F.

## 4  EXPERIMENTS

We evaluated our framework on four public image datasets, including MNIST (LeCun et al., 1998), Fashion-MNIST (Xiao et al., 2017), Small-Norb (LeCun et al., 2004) and CIFAR-10 (Krizhevsky, 2009), as well as one collection of ECG heartbeat categorization time-series data named Arrhythmia (Fazeli, 2018). Brief descriptions of these data sets can be found in Appendix G. The detail of $\beta$-VAE architecture is given in Table 3 in appendices. The number of epochs was set to 20 for experiments on MNIST and Fashion-MNIST datasets, and 50 for CIFAR-10, Small-Norb, and Arrhythmia datasets; batch size was set to 100 for experiments on all these datasets. When using t-SNE, the dimension of t-SNE map was set as 2 for all datasets, perplexity 30, the learning rate 200, and maximum number of iterations 1000. The value of $\alpha$ in the anomaly score function is searched from set $\{0.0, 0.005, 0.01, 0.05, 0.1, 0.2, 0.3, 0.4, 0.5, 0.6, 0.7, 0.8, 0.9, 0.95, 0.99, 1.0\}$. We used $k = 1$ for calculating $k$-nearest-neighbors distance in t-SNE map (we also tried to set $k$ to 3 or 5, the results were similar).

### 4.1  COMPARISON WITH CAPSNET, GANS, AND VAE

We compared AnoDM with state-of-the-art algorithms including a supervised method – CapsNet (Li et al., 2019), and two types of generative models – GANs (including AnoGAN and ADGAN) (Deecke et al., 2019) and $\beta$-VAE (implemented by setting $\alpha = 1$ thus using only reconstruction error as anomaly score). As shown in Table 1, on average, AnoDM achieved either comparable (on MNIST) or better (on Fashion-MNIST and CIFAR-10) performance in terms of receiver operating characteristic curve (auROC). On Fashion-MNIST, CapsNet (prediction-probability-based), as the best benchmark method, obtained an average auROC of 0.765, while AnoDM achieved 0.883. On MNIST, both AnoDM and CapsNet obtained the highest performance. However, CapsNet is a supervised method that takes advantages of class information, while ours is completely unsupervised that is more suitable in many practices as class information is either incomplete or unavailable. Furthermore, by comparing AnoDM with $\beta$-VAE that only considers reconstruction error as anomaly score, AnoDM dramatically improved the performance in all cases. In other words, t-SNE makes a prominent contribution to improve $\beta$-VAE for anomaly detection problems. However, all generative models did not work well on Small-Norb, mainly because these models used convolution to extract features from image, but convolution is only able to capture translation but not other affine transformations. Although CapsNet learns these transformations as a supervised method, it worth exploring unsupervised learning of affine transformations as a future topic.

Table 1: Performance of AnoDM in comparison with other methods in terms of auROC on image data. The results for AnoDM were obtained by either the $\mu$-based or $z$-based algorithm. In the column for $\beta$-VAE, only reconstruction error is used as anomaly score. The results for AnoGAN and ADGAN were obtained from (Deecke et al., 2019). The results for CapsNet were obtained from (Li et al., 2019).

| Dataset | Class | CapsNet | | AnoGAN | ADGAN | $\beta$-VAE | AnoDM |
|---|---|---|---|---|---|---|---|
| | | PP-based | RE-based | | | | |
| | 0 | 0.998 | 0.947 | 0.990 | **0.999** | 0.890 | 0.985 |
| | 1 | 0.990 | 0.907 | **0.998** | 0.992 | 0.841 | 0.987 |
| | 2 | 0.984 | 0.970 | 0.888 | 0.968 | 0.967 | **0.991** |
| | 3 | **0.976** | 0.949 | 0.913 | 0.953 | 0.947 | 0.969 |
| | 4 | 0.935 | 0.872 | 0.944 | 0.960 | 0.968 | **0.975** |
| MNIST | 5 | 0.970 | 0.966 | 0.912 | 0.955 | 0.966 | **0.976** |
| | 6 | 0.942 | 0.909 | 0.925 | 0.980 | 0.907 | **0.983** |
| | 7 | **0.987** | 0.934 | 0.964 | 0.950 | 0.899 | 0.977 |
| | 8 | **0.993** | 0.929 | 0.883 | 0.959 | 0.946 | 0.982 |
| | 9 | **0.990** | 0.871 | 0.958 | 0.965 | 0.794 | 0.928 |
| | avg. | **0.977** | 0.925 | 0.937 | 0.968 | 0.913 | 0.975 |
| | 0 | 0.620 | 0.454 | – | – | 0.500 | **0.844** |
| | 1 | 0.851 | 0.871 | – | – | 0.860 | **0.978** |
| | 2 | **0.818** | 0.486 | – | – | 0.459 | 0.783 |
| | 3 | **0.895** | 0.693 | – | – | 0.730 | 0.886 |
| | 4 | **0.790** | 0.394 | – | – | 0.379 | 0.763 |
| Fashion-MNIST | 5 | 0.691 | 0.982 | – | – | 0.985 | **0.990** |
| | 6 | **0.801** | 0.480 | – | – | 0.501 | 0.713 |
| | 7 | 0.619 | 0.787 | – | – | 0.842 | **0.952** |
| | 8 | 0.912 | 0.885 | – | – | 0.876 | **0.980** |
| | 9 | 0.656 | 0.754 | – | – | 0.701 | **0.944** |
| | avg. | 0.765 | 0.679 | – | – | 0.683 | **0.883** |
| | 0 | 0.622 | 0.371 | 0.610 | **0.661** | 0.368 | 0.635 |
| | 1 | 0.455 | 0.737 | 0.565 | 0.435 | 0.746 | **0.754** |
| | 2 | **0.671** | 0.421 | 0.648 | 0.636 | 0.397 | 0.589 |
| | 3 | **0.675** | 0.588 | 0.528 | 0.488 | 0.604 | 0.608 |
| | 4 | 0.683 | 0.388 | 0.670 | **0.794** | 0.387 | 0.564 |
| CIFAR-10 | 5 | 0.635 | 0.601 | 0.592 | **0.640** | 0.611 | 0.638 |
| | 6 | **0.727** | 0.491 | 0.625 | 0.685 | 0.500 | 0.600 |
| | 7 | **0.673** | 0.631 | 0.576 | 0.559 | 0.614 | 0.648 |
| | 8 | 0.513 | 0.410 | 0.723 | **0.798** | 0.399 | 0.642 |
| | 9 | 0.466 | 0.671 | 0.582 | 0.643 | 0.698 | **0.718** |
| | avg. | 0.612 | 0.531 | 0.612 | 0.634 | 0.532 | **0.640** |

## 4.2 IMPACT OF BETA TO PERFORMANCE

As discussed in Higgins et al. (2017), $\beta$ ($> 1$) functions as a controller to encourage most efficient latent representation learning via limiting the capacity of latent information channel. Mathieu et al. (2019) however interpreted the objective of $\beta$ as tuning a proper level of overlap of encodings by working with another term that regularizes the divergence of the aggregate posterior $q_\phi(z)$ and the desired prior $p(z)$. The main intuition is that purely increasing $\beta$ induces too much overlap which actually discourages the disentanglement of data (information which is necessary for expressing desired structure is lost). Higgins et al. (2017) demonstrated that $\beta$-VAE with $\beta > 1$ leads to interesting results when learning interpretable factorized latent representations on a variety of datasets. Surprisingly, our investigation demonstrates that by setting $0 < \beta < 1$, it actually achieved the state-of-the-art results for anomaly detection problems on a range of datasets, such as MNIST, Fashion-MNIST and Arrhythmia. Figure 2 illustrates the impact of values of $\beta$ and $\alpha$ in our anomaly score function. Interestingly, best performances were achieved when $\beta < 1$ and the performances generally degrade as $\beta$ increases, resonating with Mathieu et al. (2019) that overly large values of $\beta$ actually causes a mismatch between $q_\phi(z)$ and $p(z)$ (resulting in inappropriate level of overlap in the latent space). This phenomenon can be further seen in the t-SNE maps of latent embeddings in Figure 3. When $\beta$ becomes extremely larger than the appropriate value, the anomalous class becomes entangled with its normal neighboring classes and the boundaries between normal classes become unclear. Theoretically, adding the divergence of the aggregate posterior $q_\phi(z)$ and the desired structured prior $p(z)$ is an effective way to limit the level of overlap when $\beta$ is too large. However, it is practically challenging to design an appropriate structured prior. Therefore, in our investigation, we focused on exploring the full range of $\beta$'s value in $\beta$-VAE for the impact of disentanglement to anomaly

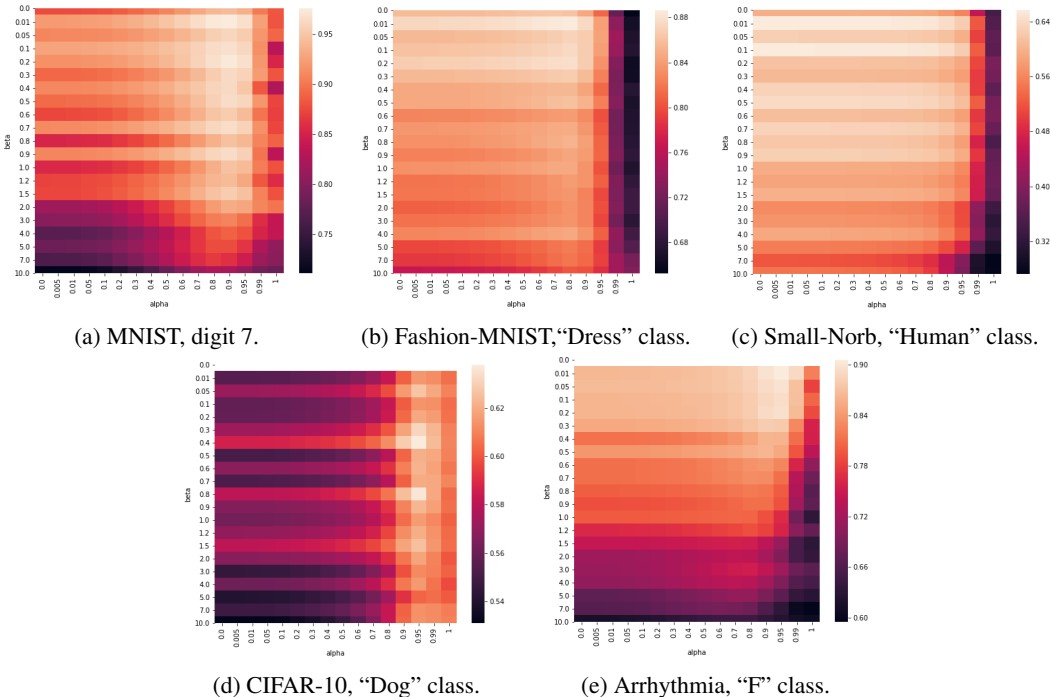

(a) MNIST, digit 7.  (b) Fashion-MNIST,"Dress" class.  (c) Small-Norb, "Human" class.

(d) CIFAR-10, "Dog" class.  (e) Arrhythmia, "F" class.

Figure 2: Performances (measured in terms of auROC) of AnoDM evaluated on five datasets: MNIST, Fashion-MNIST, Small-Norb, CIFAR-10 and Arrhythmia. On each dataset, the anomalous class is indicated in the corresponding subcaption while treating the rest classes as normal classes. Note that $\beta = 0$ doesn't work for CIFAR10 and Arrhythmia, because it makes learning highly unstable. As displayed in (d) and (e), missing values were indicated in white color at the top of corresponding heatmaps for $\beta = 0$.

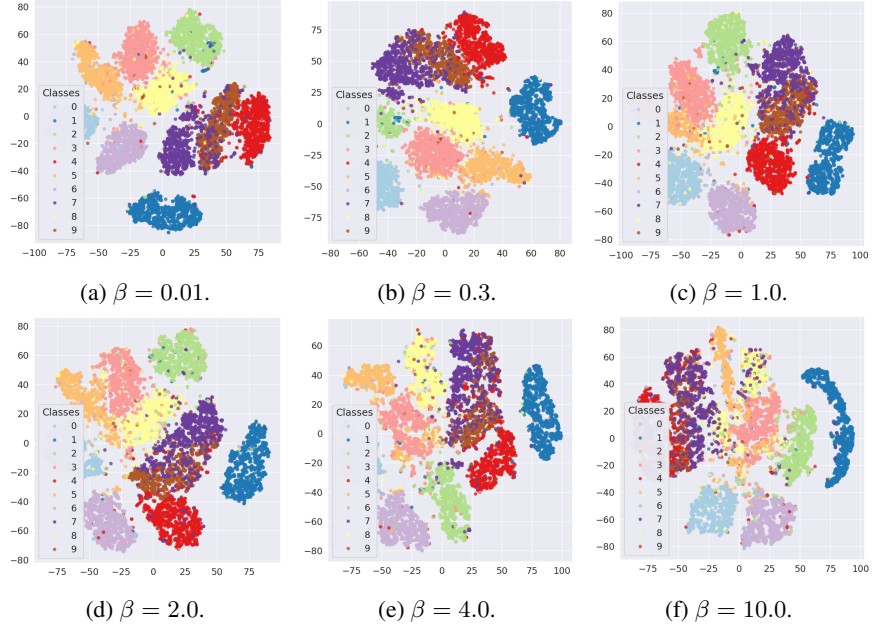

(a) $\beta = 0.01$.  (b) $\beta = 0.3$.  (c) $\beta = 1.0$.

(d) $\beta = 2.0$.  (e) $\beta = 4.0$.  (f) $\beta = 10.0$.

Figure 3: The impact of $\beta$'s value to t-SNE map of latent representations of MNIST samples. Class 7 is treated as anomalous class data. Each map displays 10000 data points identical in all maps including 5000 training data points and 5000 test data points.

detection. Since our framework is quite general, it can be easily extended to other unsupervised disentangled representation learning models for anomaly detection.

### 4.3 Impact of Beta to t-SNE Representations

The t-SNE plots in Figure 3 reflect the impact of $\beta$'s value on latent representations in the case of identifying anomalous digit 7 from MNIST. In this example, the best performance (auROC = 0.975) was achieved when $\beta = 0.01$. Clearly, as $\beta$ increases, all latent clusters become less dense, and more anomalous latent data points move to neighboring clusters. Furthermore, Figure 3 also corroborates that, even though in t-SNE maps distances between clusters might not reflect global geometry and cluster sizes might not mirror the true sizes (Wattenberg et al., 2016), using averaged distance from a test sample to its $k$ nearest normal data points represented in t-SNE space to qualify outlierness still is a very effective way for distinguishing anomalous samples when $\beta$ is tuned properly.

### 4.4 Evaluation of Anomaly Score Function

In order to better evaluate our anomaly score function, as formulated in Equation (2), we conducted a comprehensive comparison with methods only based on either distance in t-SNE map ($\mathcal{D}^k_{tSNE}$) or reconstruction-error in raw feature space ($\mathcal{D}_{RE}$). To see the contribution of t-SNE, it is also compared with the method that calculates nearest neighbor distance directly in latent space of $\beta$-VAE. Figure 4 displays the ROC curves of these four approaches when assuming anomalous classes are respectively 1 ("Trouser/pants"), 3 ("Dress"), 5 ("Sandal"), and 7 ("Sneaker") on Fashion-MNIST. It is obvious that AnoDM achieves best results among them by taking advantages of both $\beta$-VAE reconstruction and t-SNE embedding. The $\beta$-VAE reconstruction reflects whether useful information is captured by the model through recovering the input $\boldsymbol{x}$; the t-SNE embedding indicates the disentanglement of latent representations $\boldsymbol{z}$. Both measures effectively complement each other. Besides, comparing the auROCs between t-SNE-based and latent-distance-based score functions, one can clearly see that the former dramatically outperforms the latter. Same conclusion can be drawn for MNIST, CIFAR-10, and Arrhythmia, as displayed in Figures 12, 14, 16, 17, and 18 in appendices. To further show that the optimal values of $\alpha$ are close to 1 (see Figure 2) is due to magnitude difference rather than less usefulness of t-SNE, we replaced the distance score (Equation (4)) with normalized distance score in the weighted final anomaly score function (Equation (2)). We found that the optimal values of $\alpha$ shift to the lower end of the spectrum (see Figure 8). It implies that t-SNE does play a critical role in our framework.

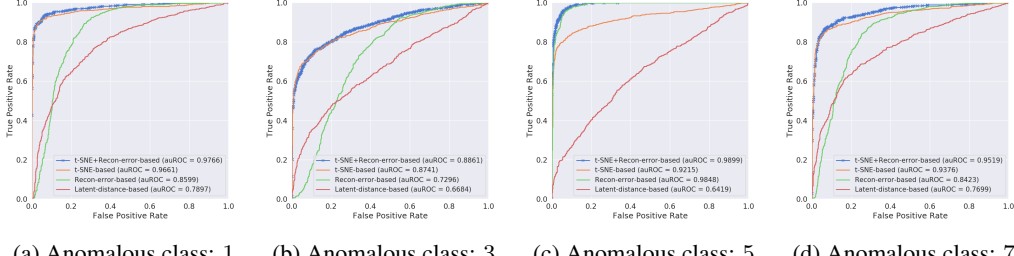

(a) Anomalous class: 1.  (b) Anomalous class: 3.  (c) Anomalous class: 5.  (d) Anomalous class: 7.

Figure 4: ROC curves of four methods on Fashion-MNIST. These four examples illustrate the results of using four different anomaly score functions: t-SNE+Recon-error-based ($S_{\beta VAE+tSNE} = \alpha \mathcal{D}_{RE} + (1 - \alpha)\mathcal{D}^k_{tSNE}$), t-SNE-based ($\mathcal{D}^k_{tSNE}$), Reconstruction-error-based ($\mathcal{D}_{RE}$) and latent-distance-based (calculating distances in latent space of $\beta$-VAE). The $\alpha$ values for these four plots are 0.8, 0.8, 0.95, and 0.8, respectively.

### 4.5 AnoDM for Time-Series

As mentioned in Section 3, our method uses a TCN encoder in $\beta$-VAE for time-series anomaly detection. Figure 5 displays the comparison among TCN, CNN and LSTM encoders in the AnoDM framework on Arrhythmia. As a special case of LSTM-$\beta$-VAE, LSTM-VAE was presented in (Park et al., 2017) for state-of-the-art sequence modelling. For the five classes in Arrhythmia, iteratively one class was treated as anomalous class, while the other classes were used as normal classes. The TCN-encoder-based method outperforms the other two methods significantly in all five cases. Even though the CNN encoder achieved impressive results when detecting anomalous class "S", "V", "F" and "Q" respectively, it did not work quite well when class "N" was treated as anomaly. One

possible reason might be that comparing with TCN and LSTM, the performance of CNN is more sensitive on the training sample size. Taking the above case as an example, as class 0 ("N") accounts for over 80% of training data, when considering it as anomaly, normal training data hence become less sufficient for learning $\beta$-VAE. Nevertheless, in TCN-based $\beta$-VAE, each hidden unit of the last deterministic hidden layer before latent encoding at the bottleneck is calculated based on much longer sequence dependency, such that it is less sensitive to the limitation of small sample size. Conclusively, as mentioned in (Bai et al., 2018), TCN should be regarded as a natural starting point for sequence modeling tasks.

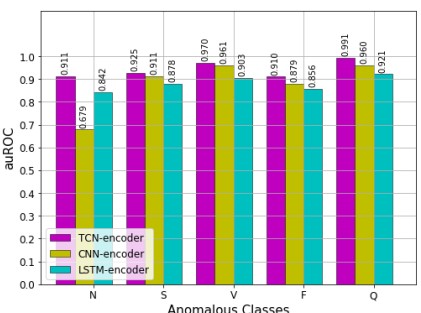

Figure 5: AuROCs on Arrhythmia.

As a case study, Figure 6a shows the original EGG signals and reconstructed signals by TCN-based $\beta$-VAE when considering class "Q" as anomaly. Normal samples can be reconstructed very well, whereas abnormal samples suffer from larger reconstruction errors. Meanwhile, the corresponding t-SNE plot in Figure 6b displays two distinctive clusters of abnormal samples. The combination of these two measures thus leads to the best performance as seen in Figure 6c.

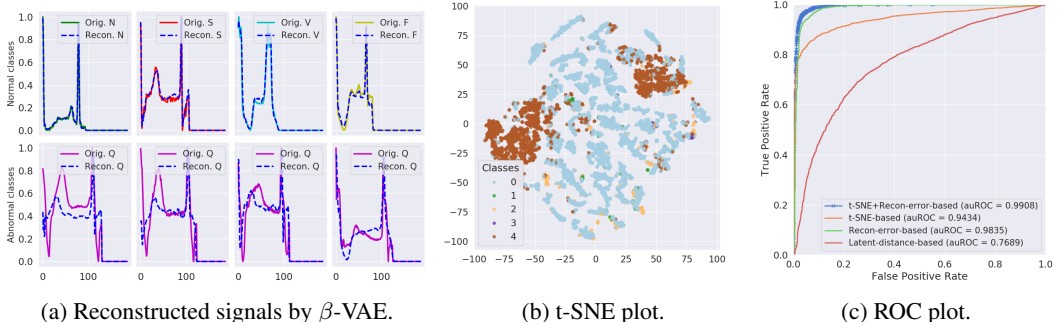

(a) Reconstructed signals by $\beta$-VAE.          (b) t-SNE plot.          (c) ROC plot.

Figure 6: Reconstructed signals, t-SNE map, and ROC curves on Arrhythmia with class "Q" as anomaly.

## 5 CONCLUSIONS

We propose a new methodology which successfully integrates t-SNE with disentangled representation learning for anomaly detection. This approach achieved state-of-the-art performances on both image data (MNIST, Fashion-MNIST and CIFAR-10) and Arrhythmia time-series data. Specifically, best performance is accomplished when $0 < \beta < 1$ for almost all cases involving $\beta$-VAE. We also defined an anomaly score function by effectively taking the advantages of both low-dimensional t-SNE embedding and $\beta$-VAE reconstruction. Our algorithm demonstrated that t-SNE plays an essential role for measuring abnormality. This research initiates the research on anomaly detection using unsupervised disentangled representation learning and lower-dimensional manifold learning. Besides, our model uses TCN network as encoding architecture for detecting anomalous time-series data and the experimental results convince us that TCN consistently outperforms CNN and LSTM. As a proof of concept, our current framework automatically inheres advantages of deep learning to address anomaly detection's issues in representability and scalability as discussed in the beginning of this paper. The extension of our framework to multimodal data is straightforward. It is also possible that a neural t-SNE component could be designed and integrated into the learning of $\beta$-VAE to achieve real-time efficiency. Other new well-performing manifold learning methods, such as UMAP (McInnes et al., 2018) which is faster and keeps global topologies, could be employed as replacement of $t$-SNE.

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

## A   Variational Lower Bound and Variational Autoencoder

A directed generative model concerns the joint distribution $p_\theta(\boldsymbol{x}, \boldsymbol{z})$ where $\boldsymbol{x}$ is a random vector variable representing an observation in $d$-dimension space, $\boldsymbol{z}$ is the vector of latent variables in $k$-dimensional space, and $\theta$ is the set of model hyperparameters. Inference $p_\theta(\boldsymbol{z}|\boldsymbol{x}) = \frac{p(\boldsymbol{x}|\boldsymbol{z})p(\boldsymbol{z})}{p(\boldsymbol{x})}$ is generally intractable, with few exceptions such as exponential family RBM (Li & Zhu, 2018b). An easy parametric distribution $q_\phi(\boldsymbol{z}|\boldsymbol{x})$ is introduced to approximate the intractable posterior $p_\theta(\boldsymbol{z}|\boldsymbol{x})$. Distribution $q_\phi(\boldsymbol{z}|\boldsymbol{x})$ should be as close to $p_\theta(\boldsymbol{z}|\boldsymbol{x})$ as possible by estimating inference parameters $\phi$. Based on mean-field variational Bayes, reverse Kullback-Leibler divergence is employed to measure the distance between these two distributions, $D_{\mathrm{KL}}\big(q_\phi(\boldsymbol{z}|\boldsymbol{x})||p_\theta(\boldsymbol{z}|\boldsymbol{x})\big)$, with respect to $\phi$. After being expanded, the following formula is obtained ($p_\theta$ is fixed with respect to $q_\phi$),

$$D_{\mathrm{KL}}\big(q_\phi(\boldsymbol{z}|\boldsymbol{x})||p_\theta(\boldsymbol{z}|\boldsymbol{x})\big) = \log p_\theta(\boldsymbol{x}) + D_{\mathrm{KL}}\big(q_\phi(\boldsymbol{z}|\boldsymbol{x})||p_\theta(\boldsymbol{z})\big) - \mathbb{E}_{\boldsymbol{z}\sim q_\phi(\boldsymbol{z}|\boldsymbol{x})}[\log p_\theta(\boldsymbol{x}|\boldsymbol{z})]. \quad (5)$$

After being rearranged, we can obtain an objective with respect to both generative $\theta$ and inference parameter $\phi$:

$$\mathcal{L}(\theta, \phi) = \log p_\theta(\boldsymbol{x}) - D_{\mathrm{KL}}\big(q_\phi(\boldsymbol{z}|\boldsymbol{x})||p_\theta(\boldsymbol{z}|\boldsymbol{x})\big) \tag{6}$$

$$= \mathbb{E}_{\boldsymbol{z}\sim q_\phi(\boldsymbol{z}|\boldsymbol{x})}[\log p_\theta(\boldsymbol{x}, \boldsymbol{z}) - \log q_\phi(\boldsymbol{z}|\boldsymbol{x})] \tag{7}$$

$$= \mathbb{E}_{\boldsymbol{z}\sim q_\phi(\boldsymbol{z}|\boldsymbol{x})}[\log p_\theta(\boldsymbol{x}|\boldsymbol{z})] - D_{\mathrm{KL}}\big(q_\phi(\boldsymbol{z}|\boldsymbol{x})||p_\theta(\boldsymbol{z})\big). \tag{8}$$

Since the KL divergence in Equation (6) is always non-negative and thus $\mathcal{L}$ is the varitional lower bound (or evidence lower bound, ELBO) of $\log p_\theta(\boldsymbol{x})$:

$$\mathcal{L} = \log p_\theta(\boldsymbol{x}) - D_{\mathrm{KL}}\big(q_\phi(\boldsymbol{z}|\boldsymbol{x})||p_\theta(\boldsymbol{z}|\boldsymbol{x})\big) \leq \log p_\theta(\boldsymbol{x}). \tag{9}$$

Equation (8) implies that maximizing the ELBO is to maximize the reconstruction error and also minimize the difference between the inference distribution and generative prior distribution. From the variational Bayes perspective, the aim is to obtain the optimal generative and inference parameters through maximizing the ELBO:

$$\hat{\theta}, \hat{\phi} = \underset{\theta, \phi}{\operatorname{argmax}} \mathcal{L}. \tag{10}$$

The main challenge in learning of directed generative models is to derive the gradient of Equation (7) with respect to the inference parameter. Therefore, the reparameterization trick is introduced in VAE (Kingma & Welling, 2014; Rezende et al., 2014) such that backpropagation can be used through expressing the random variable $\boldsymbol{z}$ as a deterministic variable $\boldsymbol{z} = \tau_\phi(\boldsymbol{x}, \epsilon)$, where $\epsilon$ is an auxiliary independent random variable, and the transformation function $\tau_\phi$ parameterized by $\phi$ converts $\epsilon$ to $\boldsymbol{z}$. For example, when $q_\phi(\boldsymbol{z}|\boldsymbol{x})$ is a multivariate Gaussian with a diagonal covariance structure ($q_\phi(\boldsymbol{z}|\boldsymbol{x}) = \mathcal{N}(\boldsymbol{z}; \boldsymbol{\mu}, \boldsymbol{\sigma}^2 \boldsymbol{I})$), sampling from this distribution is equivalent to:

$$\epsilon \sim \mathcal{N}(\boldsymbol{0}, \boldsymbol{I}), \tag{11}$$

$$\boldsymbol{z} = \boldsymbol{\mu}(\boldsymbol{x}) + \boldsymbol{\sigma}(\boldsymbol{x}) \odot \epsilon, \tag{12}$$

where $[\boldsymbol{\mu}(\boldsymbol{x}), \boldsymbol{\sigma}(\boldsymbol{x})] = f_\phi(\boldsymbol{x})$ is a deterministic two-head multilayer perceptrons parameterized by $\phi$, and $\odot$ denotes element-wise product. The reparametrization trick enables $\mathbb{E}_{\boldsymbol{z}\sim q(\boldsymbol{z}|\boldsymbol{x})}[\log p_\theta(\boldsymbol{x}, \boldsymbol{z}) - \log q_\phi(\boldsymbol{z}|\boldsymbol{x})] = \mathbb{E}_\epsilon[\log p_\theta(\boldsymbol{x}, \boldsymbol{z}) - \log q_\phi(\boldsymbol{z}|\boldsymbol{x})]$, such that the gradient of Equation (7) with respect to $\phi$ can be reformulated to

$$\frac{\partial \mathcal{L}(\theta, \phi)}{\partial \phi} = \frac{\partial \mathbb{E}_{\boldsymbol{z}\sim q(\boldsymbol{z}|\boldsymbol{x})}[\log p_\theta(\boldsymbol{x}, \boldsymbol{z}) - \log q_\phi(\boldsymbol{z}|\boldsymbol{x})]}{\partial \phi} \tag{13}$$

$$= \frac{\partial \mathbb{E}_\epsilon[\log p_\theta(\boldsymbol{x}, \boldsymbol{z}) - \log q_\phi(\boldsymbol{z}|\boldsymbol{x})]}{\partial \phi} \tag{14}$$

$$= \mathbb{E}_\epsilon \left[ \frac{\partial \big(\log p_\theta(\boldsymbol{x}, \boldsymbol{z}) - \log q_\phi(\boldsymbol{z}|\boldsymbol{x})\big)}{\partial \phi} \right] \tag{15}$$

$$= \mathbb{E}_\epsilon \left[ \frac{\partial \big(\log p_\theta(\boldsymbol{x}, \tau_\phi(\boldsymbol{x}, \epsilon)) - \log q_\phi(\boldsymbol{z}|\boldsymbol{x})\big)}{\partial \phi} \right]. \tag{16}$$

The gradient with respect to generative parameters $\theta$ is easier, because it does not rely on the reparameterization trick:

$$\frac{\partial \mathcal{L}(\theta, \phi)}{\partial \theta} = \frac{\partial \, \mathbb{E}_{\boldsymbol{z} \sim q(\boldsymbol{z}|\boldsymbol{x})}[\log p_\theta(\boldsymbol{x}, \boldsymbol{z}) - \log q_\phi(\boldsymbol{z}|\boldsymbol{x})]}{\partial \theta} \tag{17}$$

$$= \mathbb{E}_{\boldsymbol{z} \sim q(\boldsymbol{z}|\boldsymbol{x})} \left[ \frac{\partial \big( \log p_\theta(\boldsymbol{x}, \boldsymbol{z}) - \log q_\phi(\boldsymbol{z}|\boldsymbol{x}) \big)}{\partial \theta} \right] \tag{18}$$

$$= \mathbb{E}_{\boldsymbol{z} \sim q(\boldsymbol{z}|\boldsymbol{x})} \left[ \frac{\partial \log p_\theta(\boldsymbol{x}, \boldsymbol{z})}{\partial \theta} \right]. \tag{19}$$

# B   BETA-VAE AND BEYOND FOR DISENTANGLED REPRESENTATION LEARNING

Higgins et al. (2017) proposed a novel deep generative model, named $\beta$-VAE, a modification of VAE by introducing an adjustable hyperparameter $\beta$ to learn an interpretable disentangled representation of the data generative latent factors. Specifically, $\beta$ functions as a controller to trade off between the extent of learning constraints and reconstruction accuracy. The constraints impose a limit on the capacity of the latent information channel and an emphasis on learning statistically independent latent factors. Higgins et al. (2017) demonstrated that $\beta$-VAE with appropriately tuned $\beta$ ($\beta > 1$) qualitatively outperforms VAE ($\beta = 1$) as well as state of the art unsupervised (InfoGAN) and semi-supervised (DC-IGN) approaches to disentangled factor learning on a variety of datasets (celebA, faces and chairs).

Higgins et al. (2017) assumed that an image $\boldsymbol{x}$ is generated by the true world simulator using ground truth data generative factors: $p(\boldsymbol{x}|\boldsymbol{v}, \boldsymbol{w}) = \text{Sim}(\boldsymbol{v}, \boldsymbol{w})$, where $\boldsymbol{v}$ is set of conditionally independent factors and $\boldsymbol{w}$ is set of conditionally dependent factors. Therefore, the joint distribution of the data $\boldsymbol{x}$ and a set of generative latent factors $\boldsymbol{z}$ is: $p(\boldsymbol{x}|\boldsymbol{z}) \approx p(\boldsymbol{x}|\boldsymbol{v}, \boldsymbol{w}) = \text{Sim}(\boldsymbol{v}, \boldsymbol{w})$. The aim of this generative model is then to ensure that the inferred latent factors from $q_\phi(\boldsymbol{z}|\boldsymbol{x})$ capture the generative factors $\boldsymbol{v}$ in a disentangled manner. The conditionally dependent data generative factors $\boldsymbol{w}$ can remain entangled in a separate subset of $\boldsymbol{z}$ that is not entangled with $\boldsymbol{v}$ . Considering the prior $p(\boldsymbol{z})$ is set to be an isotropic unit Gaussian $p(\boldsymbol{z}) = \mathcal{N}(\boldsymbol{0}, \boldsymbol{I})$, a constraint $\delta$ is introduced to encourage the matching between $q_\phi(\boldsymbol{z}|\boldsymbol{x})$ and $p(\boldsymbol{z})$ such that the disentangling property in the inferred $q_\phi(\boldsymbol{z}|\boldsymbol{x})$ can be realized.

Following the same incentive as in VAE: maximizing the probability of generating real data, while minimizing the distance between the generative and approximate posterior distributions, as formulated below

$$\max_{\phi, \theta} \mathbb{E}_{\boldsymbol{x} \sim \mathbb{X}} \big[ \, \mathbb{E}_{q_\phi(\boldsymbol{z}|\boldsymbol{x})}[\log p_\theta(\boldsymbol{x}|\boldsymbol{z})] \big] \tag{20}$$

$$\text{s.t. } D_{\text{KL}} \big( q_\phi(\boldsymbol{z}|\boldsymbol{x}) || p(\boldsymbol{z}) \big) < \delta, \tag{21}$$

where $\mathbb{X} = \{\boldsymbol{x}_1, \boldsymbol{x}_2, \ldots, \boldsymbol{x}_n\}$ is the training data set. Rewriting this equation as a Lagrangian under the KKT conditions (Kuhn & Tucke, 1951; Karush, 1939), the following equation $\mathcal{F}(\theta, \phi, \beta)$ is obtained:

$$\mathcal{F}(\theta, \phi, \beta)$$
$$= \mathbb{E}_{q_\phi(\boldsymbol{z}|\boldsymbol{x})}[\log p_\theta(\boldsymbol{x}|\boldsymbol{z})] - \beta(D_{\text{KL}} \big( q_\phi(\boldsymbol{z}|\boldsymbol{x}) || p(\boldsymbol{z})) - \delta)$$
$$= \mathbb{E}_{q_\phi(\boldsymbol{z}|\boldsymbol{x})}[\log p_\theta(\boldsymbol{x}|\boldsymbol{z})] - \beta D_{\text{KL}} \big( q_\phi(\boldsymbol{z}|\boldsymbol{x}) || p(\boldsymbol{z})) - \beta \delta$$
$$\geq \mathbb{E}_{q_\phi(\boldsymbol{z}|\boldsymbol{x})}[\log p_\theta(\boldsymbol{x}|\boldsymbol{z})] - \beta D_{\text{KL}} \big( q_\phi(\boldsymbol{z}|\boldsymbol{x}) || p(\boldsymbol{z})). \tag{22}$$

The objective function to be maximized in $\beta$-VAE is thus defined as:

$$\mathcal{L}_\beta(\theta, \phi) = \mathbb{E}_{q_\phi(\boldsymbol{z}|\boldsymbol{x})}[\log p_\theta(\boldsymbol{x}|\boldsymbol{z})] - \beta D_{\text{KL}} \big( q_\phi(\boldsymbol{z}|\boldsymbol{x}) || p(\boldsymbol{z})), \tag{23}$$

where the Lagrangian multiplier $\beta$ is the regularisation coefficient that constrains the capacity of the latent information channel $\boldsymbol{z}$ and puts implicit independence pressure on the learnt posterior due to the isotropic nature of the Gaussian prior $p(\boldsymbol{z})$. When $\beta = 1$, $\beta$-VAE corresponds to the original VAE formulation of Kingma & Welling (2014). When $\beta > 1$, it applies stronger constraint which limits the capacity of $\boldsymbol{z}$ and encourages the model to learn the most efficient representation of the

data. Theoretically, a higher $\beta$ encourages more efficient latent encoding and further encourages the disentanglement. However, a higher $\beta$ may lead to poorer reconstructions due to the loss of high frequency details when passing through a constrained latent bottleneck.

Burgess et al. (2018) proposed an improvement to $\beta$-VAE, by progressively increasing the information capacity of the latent code during training. It facilitates the robust learning of disentangled representations in $\beta$-VAE without the previous trade-off in reconstruction accuracy. The objective function to be maximized in $\beta$-VAE is redefined as:

$$\mathcal{L}_\beta(\theta, \phi) = \mathbb{E}_{q_\phi(\boldsymbol{z}|\boldsymbol{x})}[\log p_\theta(\boldsymbol{x}|\boldsymbol{z})] - \beta |D_{KL}(q_\phi(\boldsymbol{z}|\boldsymbol{x})||p(\boldsymbol{z})) - C|, \tag{24}$$

where the hyperparameter $\beta$ controls how heavily to penalise the deviation of $D_{\mathrm{KL}}\big(q_\phi(\boldsymbol{z}|\boldsymbol{x})||p(\boldsymbol{z})\big)$ and a controllable value $C$. By gradually increasing $C$ from zero to a large value, good-quality reconstruction can be obtained.

Hoffman et al. (2017) had also made a deeper research on $\beta$-VAE considering $\beta < 1$. They argued that optimizing this partially regularized ELBO is equivalent to performing variational expectation maximization (EM) with an implicit prior $r(\boldsymbol{z})$ ($r(\boldsymbol{z}) \propto q_\phi(\boldsymbol{z})^{(1-\beta)}p(\boldsymbol{z})^\beta$) that depends on the marginal (aggregate) posterior $q(\boldsymbol{z}) \triangleq \frac{1}{N}\sum_{i=1}^{N} q(\boldsymbol{z}|\boldsymbol{x}^{(i)})$, and further derived some approximations to examine this prior.

Mathieu et al. (2019) explained that most recent work for learning disentangled representations of data with deep generative models has focused on capturing purely independent factors of variation by employing regularizers explicitly encouraging independence in the representations. They argued that such an approach is not generalisable, and quite restrictive for complex models where the true generative factors are not independent, very large in number, or where a set of true generative factors cannot be well defined. Overly large $\beta$ is not universally beneficial for disentanglement, Since this in turn causes a mismatch between marginal posterior $q_\phi(\boldsymbol{z})$ and the prior $p(\boldsymbol{z})$. Thus they proposed a generalization of disentanglement in VAE by explicitly separating such a decomposition as two tasks: a) the latent encoding of data should achieved an appropriate level of non-negligible overlap in aggregate encoding $q_\phi(\boldsymbol{z})$, and b) the aggregate encoding of data $q_\phi(\boldsymbol{z})$ should match the prior $p(\boldsymbol{z})$ which demonstrates the desired dependency structure between latent variables. By improving the match of $q_\phi(\boldsymbol{z})$ and $p(\boldsymbol{z})$, the overlap is only up to an appropriate level. Mathieu et al. (2019) developed a new objective that incorporates both a) and b) by introducing an additional divergence term $\mathbb{D}\big(q_\phi(z), p(\boldsymbol{z})\big)$. The objective of this improved $\beta$-VAE is defined as below:

$$\mathcal{L}_{\alpha,\beta}(\boldsymbol{x}) = \mathbb{E}_{q_\phi(\boldsymbol{z}|\boldsymbol{x})}[\log p_\theta(\boldsymbol{x}|\boldsymbol{z})] - \beta KL(q_\phi(\boldsymbol{z}|\boldsymbol{x})||p(\boldsymbol{z})) - \alpha \mathbb{D}(q_\phi(\boldsymbol{z}), p(\boldsymbol{z})). \tag{25}$$

By appropriately setting $\beta$ and $\alpha$, it allows direct control over the level of overlap and the regularization between the marginal posterior and the prior. However, a practical challenge of this method is how to define a proper structured prior when the structure of real hidden factors is poorly known. For this reason, our computational experiment in this paper is based on Burgess et al. (2018)'s $\beta$-VAE.

Instead of employing the regularization controller $\beta$ to disentangle the independent data factors by putting implicit independence pressure in the representation, Hamaguchi et al. (2018) presented a new method to learn disentangled representation by introducing two additional loss functions $\mathcal{L}_{\mathrm{sim}}$ (a similarity loss function) and $\mathcal{L}_{\mathrm{act}}$ (an activation function). This technique aims to detect trivial events in an image resulting from environmental changes (such as illumination changes, background motions and shadows) by disentangling each image into two kinds of features, specific and common. The functionality of $\mathcal{L}_{\mathrm{sim}}$ constrains common features to represent invariant factors between two paired images. $\mathcal{L}_{\mathrm{act}}$ encourages activation of common features to avoid a trivial solution. After common features of paired images are separated, the means of common features from two images are fed into event detector (a classifier) for training. As mentioned in Hamaguchi et al. (2018), this method cannot achieve good disentanglement when dealing with complicated scenes, since the activation of the units in the common features are degenerated a certain value.

## C    T-SNE FOR MANIFOLD LEARNING

The t-distributed stochastic neighbor embedding (t-SNE), introduced by van der Maaten & Hinton (2008), is a popular technique for nonlinear dimensionality reduction that is particularly well suited for visualizing similarity of high-dimensional data that lie on several different, but related, low-dimensional manifolds. It is an improvement of stochastic neighbor embedding (SNE) presented

by Hinton & Roweis (2002) with easier optimization and better visualization via replacing the SNE cost function with a symmetrized version and using a Student-t distribution instead of a Gaussian to compute the similarity between two points in the low-dimensional space.

## C.1 STOCHASTIC NEIGHBOR EMBEDDING (SNE)

SNE measures similarities (in both original high-dimensional space and mapped low-dimensional space) between points by converting Euclidean distances between data points into conditional probabilities under Gaussian distributions. In original high-dimensional space, for example, the similarity of a pair of data points $x_i$ and $x_j$ is represented by $p_{j|i}$ under a Gaussian centered at $x_i$. High $p_{j|i}$ means $x_i$ and $x_j$ are close to each other, and vice versa. $p_{j|i}$ is defined as:

$$p_{j|i} = \frac{\exp(-||x_i - x_j||^2/2\sigma_i^2)}{\sum_{k \neq i} \exp(-||x_i - x_k||^2/2\sigma_i^2))}, \tag{26}$$

where $\sigma_i$ is the variance of the Gaussian that is centered on data point $x_i$. It is influenced by the perplexity $\text{Perp}(P_i)$, which is defined as follows,

$$\text{Perp}(P_i) = 2^{H(P_i)}, \tag{27}$$

where $P_i$ corresponds particular $\sigma_i$ and $H(P_i)$ is the Shannon entropy of $P_i$ measured in bits.

For the low-dimensional counterparts $y_i$ and $y_j$ of the high-dimensional data points $x_i$ and $x_j$, the similarity of these two points is denoted as $q_{j|i}$, and similarly defined as:

$$q_{j|i} = \frac{\exp(-||y_i - y_j||^2)}{\sum_{k \neq i} \exp(-||y_i - y_k||^2)}, \tag{28}$$

where the variance of the Gaussian is set to $\frac{1}{\sqrt{2}}$. Since the focus is modeling pairwise similarities, both $p_{i|i}$ and $q_{i|i}$ are set to zero.

## C.2 COST FUNCTION OF T-SNE

Kullback-Leibler divergence is utilized to measure the difference between the low-dimensional data representation distribution $q_{j|i}$ and high-dimensional data distribution $p_{j|i}$. The cost function of SNE, which is sum of Kullback-Leibler divergences over all data points, to be optimized by a gradient descent method, is given by

$$C = \sum_i D_{\text{KL}}(P_i||Q_i) = \sum_i \sum_j p_{j|i} \log \frac{p_{j|i}}{q_{j|i}}. \tag{29}$$

Aiming to alleviate the difficult optimization problem of the cost function above and "crowding problem" (comparing nearby data points, moderate distant data points will not occupy reasonably large area in the low-dimensional map), van der Maaten & Hinton (2008) introduced symmetric SNE and employed a Student t-distribution with one degree of freedom (which is the same as a Cauchy distribution) as the heavy-tailed distribution in the low-dimensional map. The cost function is redefined as:

$$C = D_{\text{KL}}(P||Q) = \sum_i \sum_j p_{ij} \log \frac{p_{ij}}{q_{ij}}, \tag{30}$$

where again, $p_{ii}$ and $q_{ii}$ are also set to zero, and

$$p_{ij} = \frac{\exp(-||x_i - x_j||^2/2\sigma^2)}{\sum_{k \neq l} \exp(-||x_k - x_l||^2/2\sigma^2)}, \tag{31}$$

$$q_{ij} = \frac{(1 + ||y_i - y_j||^2)^{-1}}{\sum_{k \neq l}(1 + ||y_k - y_l||^2)^{-1}}. \tag{32}$$

The gradient of the Kullback-Leibler divergence between $P$ and the Student-t based joint probability distribution $Q$ is given by

$$\frac{\delta C}{\delta y_i} = 4 \sum_i (p_{ij} - q_{ij})(y_i - y_j)(1 + ||y_i - y_j||^2)^{-1}. \tag{33}$$

## D   TEMPORAL CONVOLUTIONAL NETWORKS

Bai et al. (2018) employed a basic architecture which is essentially same as the time delay neural network proposed by Waibel et al. (1990) to ensure outputs of same length as inputs and no leakage from the future into the past.

$$TCN = 1D\ FCN + causal\ convolutions.$$

However, since a simple causal convolutions is not able to achieve a long effective history size, Bai et al. (2018) employed dilated causal convolutions (same architecture as WaveNet (van den Oord et al., 2016), please refer to Figure 7a) that enable an exponentially large receptive field. The dilated convolution operation $F$ on element $s$ of the sequence is defined as:

$$F(s) = (\boldsymbol{x} *_d f)(s) = \sum_{i=0}^{k-1} f(i) \cdot \boldsymbol{x}_{s-d \cdot i}, \tag{34}$$

where $\boldsymbol{x} \in \mathbb{R}^n$ is a 1-D sequence input, $f : 0, \ldots, k-1 \to \mathbb{R}$ is a filter, $d$ is the dilation factor, $k$ is the filter size, and $s - d \cdot i$ accounts for the direction of the past. When $d = 1$, a dilated convolution reduces to a regular convolution. By choosing larger filter sizes $k$ and increasing the dilation factor $d$, the receptive field of TCN can be enlarged.

Bai et al. (2018) also utilized a generic residual module (He et al., 2015) in place of a convolutional layer. A residual block is defined as:

$$\text{o} = \text{Activation}(\boldsymbol{x} + \mathcal{F}(\boldsymbol{x})).$$

The outputs of a series of transformations $\mathcal{F}$ are added to the input $\boldsymbol{x}$ of the block. To tackle with discrepancy of input and output widths in a standard residual block, an additional $1 \times 1$ convolution is used to ensure that element wise addition $\oplus$ receives tensors of the same shape (see Figure 7b).

Bai et al. (2018) further discussed the advantages of TCN (including parallelism, flexible receptive field size, stable gradients, low memory requirement for training and variable length inputs) and its disadvantages (including possibly high memory requirement for evaluation and potential parameter change for a transfer of domain).

## E   THE FULL ANODM ALGORITHM

---
**Algorithm 1:** AnoDM Algorithm

---
**Result:** Anomaly scores of test samples
**Inputs:** $\boldsymbol{X}_{\text{tr}}$: training samples, $\boldsymbol{X}_{\text{te}}$: test samples, $\beta > 0$: hyperparameter for $\beta$-VAE
1  **while** *epoch no more than training iterations* **do**
2      Encoder net maps $\boldsymbol{X}_{\text{tr}}$ into $\boldsymbol{\mu}_{\text{tr}}$ and $\boldsymbol{\sigma}_{\text{tr}}$;
3      $\boldsymbol{Z}_{\text{tr}} = \boldsymbol{\mu}_{\text{tr}} + \boldsymbol{\sigma}_{\text{tr}} \odot \boldsymbol{\epsilon}, \ \boldsymbol{\epsilon} \sim \mathcal{N}(0, \mathcal{I})$;
4      Decoder net reconstructs $\boldsymbol{X}_{\text{tr}}$ to $\boldsymbol{X}'_{\text{tr}}$ using $\boldsymbol{Z}_{\text{tr}}$;
5      Update $\beta$-VAE's parameters $\boldsymbol{\theta}$ and $\boldsymbol{\phi}$;
6  **end**
7  For $\boldsymbol{X}_{\text{tr}}$ and $\boldsymbol{X}_{\text{te}}$, obtain $\boldsymbol{\mu}_{\text{tr}}$ and $\boldsymbol{\mu}_{\text{te}}$ respectively using trained $\beta$-VAE;
8  Use $t$-SNE to map $\boldsymbol{\mu}_{\text{tr}}$ and $\boldsymbol{\mu}_{\text{te}}$ to 2D representations $\boldsymbol{l}_{\text{tr}}$ and $\boldsymbol{l}_{\text{te}}$;
9  **for** $\boldsymbol{x}_{te}^{(i)}$ *within* $\boldsymbol{X}_{te}$ **do**
10      $\mathcal{D}_{\text{RE}}(\boldsymbol{x}_{\text{te}}^{(i)}) \triangleq \text{NSE}(\boldsymbol{x}_{\text{te}}^{(i)}, \boldsymbol{x}_{\text{te}}'^{(i)}) = \frac{\|\boldsymbol{x}_{\text{te}}^{(i)} - \boldsymbol{x}_{\text{te}}'^{(i)}\|_2^2}{\|\boldsymbol{x}_{\text{te}}^{(i)}\|_2}$ ;      `// reconstruction error`
11      $\mathcal{D}_{\text{tSNE}}^k(\boldsymbol{x}_{\text{te}}^{(i)}) \triangleq \frac{1}{k} \sum_{j \in N(i,k)} \|\boldsymbol{l}_{\text{te}}^{(i)} - \boldsymbol{l}_{\text{tr}}^{(j)}\|_2$ ; `// ` $N(i,k)$ ` is the set of indices of ` $\boldsymbol{l}_{\text{te}}^{(i)}$ `'s`
      `k nearest neighbors from ` $\boldsymbol{l}_{\text{tr}}$
12      $\mathcal{S}_{\beta\text{VAE+SNE}}(\boldsymbol{x}_{\text{te}}^{(i)}) = \alpha \mathcal{D}_{\text{RE}} + (1-\alpha)\mathcal{D}_{\text{tSNE}}^k$ ;      `// ` $\alpha \in [0,1]$
13  **end**

---

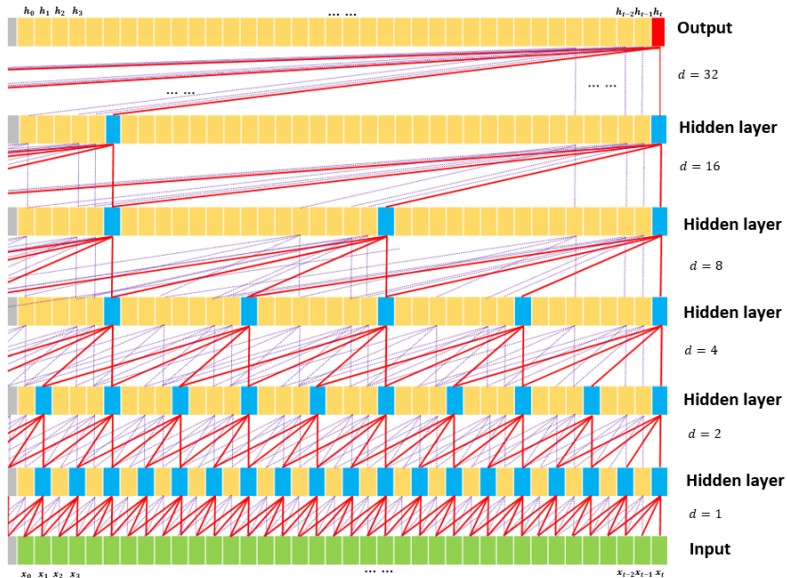

(a) A dilated causal convolution with dilation factors $d = [1, 2, 4, 8, 16, 32]$ and filter size $k = 4$.

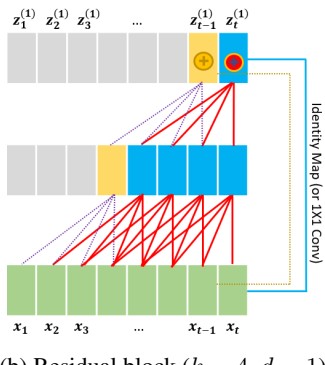

(b) Residual block ($k = 4$, $d = 1$).

Figure 7: Architecture of the temporal convolutional network (TCN).

## F    ANOMALY SCORES WITH NORMALIZED K-NN DISTANCE IN T-SNE MAPS

Since the normalised reconstruction error in input space and the $k$-NN distance in t-SNE maps may have very different magnitudes (as mentioned in Section 3.5), some values of $\alpha$ in Equation (2) are close to (but not exactly equal to) 1. The reader may have the intuition that t-SNE is not useful in AnoDM. A simple way, to find out whether large $\alpha$ value is due to magnitude difference or useless of t-SNE, is to replace $k$-NN distance of a test sample in t-SNE map (Equation (4)) with a $k$-NN distance normalized by average distance among training samples in the t-SNE map as defined below:

$$\mathcal{N}_{\text{tSNE}}^k(\boldsymbol{x}_{\text{te}}^{(i)}) \triangleq \frac{\mathcal{D}_{\text{tSNE}}^k(\boldsymbol{x}_{\text{te}}^{(i)})}{c * \mathcal{D}_{\text{tSNE}}(\boldsymbol{x}_{\text{tr}})}, \tag{35}$$

$$\mathcal{D}_{\text{tSNE}}(\boldsymbol{x}_{\text{tr}}) \triangleq \frac{1}{n} \sum_{i,j \in \{1,2,\dots,n\}} \|\boldsymbol{x}_{\text{tr}}^{(i)} - \boldsymbol{x}_{\text{tr}}^{(j)}\|_2, \tag{36}$$

where $c > 0$ is normalization hyperparameter (we set it to 0.5 in our experiment); $\boldsymbol{x}_{\text{tr}}$ is a training sample; $n$ is the total number of training samples in a t-SNE map. The heatmaps in Figure 8 depict performances of AnoDM on Fashion-MNIST ("Dress"class is considered as anomaly) and Arrhythmia ("F" class is treated as anomaly) using normalized $k$-NN distance in t-SNE maps in combination reconstruction error. By comparing Figure 8 with Figure 2, one can see that the optimal values of $\alpha$ shift upper right corner area to the upper left corner area. It thus implies that, the optimal

value of $\alpha$ is affected by the magnitude difference, and t-SNE indeed plays an essential role in AnoDM.

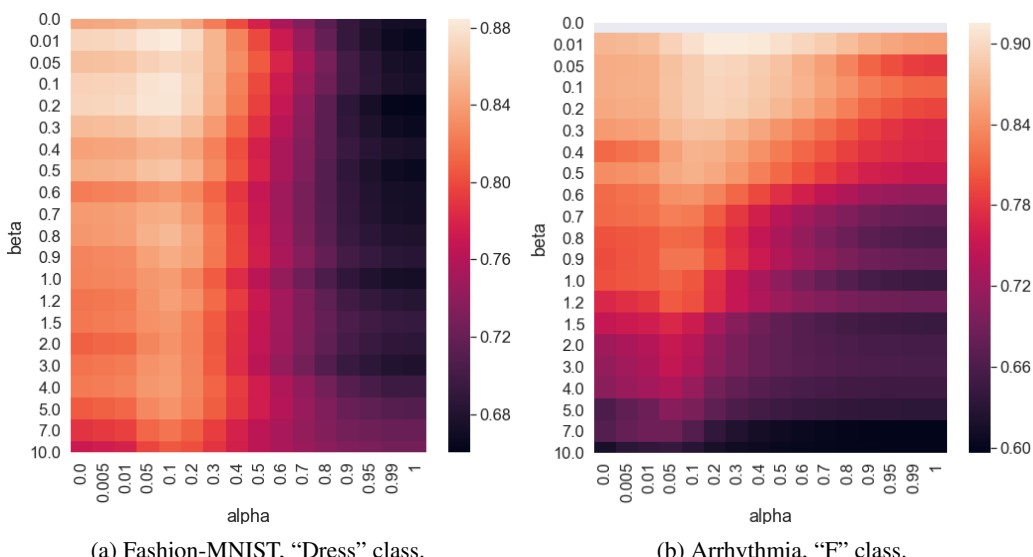

(a) Fashion-MNIST, "Dress" class.  (b) Arrhythmia, "F" class.

Figure 8: Performances (measured in terms of auROC) of AnoDM evaluated on Fashion-MNIST and Arrhythmia when using normalized $k$-NN distance in t-SNE maps in combination with reconstruction error. On each dataset, the anomalous class is indicated in the corresponding subcaption while treating the rest classes as normal classes.

## G   DESCRIPTIONS OF DATA SETS

- **MNIST**: It contains a training set of 60000 gray scale digit images of $28 \times 28$ and a test set of 10000 same resolution gray scale examples from approximately 500 different writers (LeCun et al., 1998).

- **Fashion-MNIST**: It is a dataset of Zalando's article images, comprising 70000 MNIST-like labeled fashion images of $28 \times 28$, with 7000 images per category (Xiao et al., 2017). The training set has 60000 images and the test set has 10000 images. The samples come from 10 classes: T-shirt/top, trouser, pullover, dress, coat, sandal, shirt, sneaker, bag, and ankle boot.

- **Small-Norb**: It contains 24300 $96 \times 96$ grayscale images pairs of 50 toys belonging to 5 generic categories: four-legged animals, human figures, airplanes, trucks, and cars (LeCun et al., 2004). The objects were imaged by two cameras under 6 lighting conditions, 9 elevations , and 18 azimuths. As in (Sabour et al., 2017), the images were resized to $48 \times 48$; random $32 \times 32$ crops of them were obtained during training process. Central $32 \times 32$ patches of test images were used during test.

- **CIFAR-10**: It consists of 60000 $32 \times 32$ colour images in 10 classes, with 6000 images per class. There are 10000 test images which include exactly 1000 randomly-selected images from each class and 50000 training images (with 5000 images from each class) are randomly grouped into 5 batches (Krizhevsky, 2009).

- **Arrhythmia**: It is derived from Physionet's MIT-BIH Arrhythmia Datase which consists of ECG recordings from 47 different subjects recorded at the sampling rate of $360H_z$. This dataset is created by five different heart beat categories: "N", "S", "V", "F" and "Q", in accordance with Association for the Advancement of Medical Instrumentation (AAMI) EC57 standard (Kachuee et al., 2018). The meanings of these categories are explained below.

    1. "N": normal, left or right bundle branch block, atrial escape and nodal escape.

2. "S": atrial premature, aberrant atrial premature, nodal premature and supra-ventricular premature.
3. "V": premature ventricular contraction and ventricular escape.
4. "F": fusion of ventricular and normal.
5. "Q": paced, fusion of paced and normal and unclassifiable.

Table 2: AuROCs for both $\mu$-based and $z$-based techniques.

| Dataset | Class | $\mu$-based | | | $z$-based | | |
|---|---|---|---|---|---|---|---|
| | | auROC | $\beta$ | $\alpha$ | auROC | $\beta$ | $\alpha$ |
| MNIST | 0 | 0.984 | 0.01 | 0.8 | **0.985** | 0.8 | 0.1 |
| | 1 | 0.986 | 0.01 | 0.6 | **0.987** | 0.01 | 0.6 |
| | 2 | 0.990 | 0.4 | 0.9 | **0.991** | 0.2 | 0.9 |
| | 3 | **0.969** | 0.05 | 0.9 | 0.968 | 0.05 | 0.9 |
| | 4 | 0.974 | 0.01 | 0.95 | **0.975** | 0.1 | 0.95 |
| | 5 | 0.975 | 0.01 | 0.95 | **0.976** | 0.01 | 0.95 |
| | 6 | **0.983** | 0.01 | 0.8 | 0.980 | 0.01 | 0.8 |
| | 7 | 0.975 | 0.01 | 0.9 | **0.977** | 0.01 | 0.9 |
| | 8 | 0.980 | 0.01 | 0.9 | **0.982** | 0.01 | 0.9 |
| | 9 | **0.928** | 0.4 | 0.8 | 0.925 | 0.05 | 0.8 |
| | avg. | 0.974 | – | – | **0.975** | - | – |
| Fashion-MNIST | 0 | **0.844** | 0.05 | 0.2 | 0.840 | 0.05 | 0.0 |
| | 1 | 0.977 | 0.01 | 0.8 | **0.978** | 0.01 | 0.8 |
| | 2 | **0.783** | 0.01 | 0.0 | **0.783** | 0.05 | 0.0 |
| | 3 | **0.886** | 0.01 | 0.8 | 0.884 | 0.01 | 0.8 |
| | 4 | 0.760 | 0.1 | 0.0 | **0.763** | 0.3 | 0.0 |
| | 5 | **0.990** | 0.2 | 0.95 | **0.990** | 0.2 | 0.95 |
| | 6 | **0.713** | 0.05 | 0.0 | 0.709 | 0.05 | 0.0 |
| | 7 | **0.952** | 0.01 | 0.8 | 0.950 | 0.01 | 0.8 |
| | 8 | **0.980** | 0.05 | 0.8 | **0.980** | 0.05 | 0.8 |
| | 9 | 0.940 | 0.3 | 0.8 | **0.944** | 0.3 | 0.7 |
| | avg. | **0.883** | – | – | 0.882 | - | – |
| CIFAR-10 | 0 | **0.635** | 0.4 | 0.0 | 0.634 | 0.2 | 0.0 |
| | 1 | 0.752 | 0.6 | 0.99 | **0.754** | 0.05 | 0.99 |
| | 2 | **0.589** | 1.0 | 0.0 | 0.558 | 0.7 | 0.0 |
| | 3 | **0.608** | 10.0 | 0.95 | 0.606 | 10.0 | 0.99 |
| | 4 | **0.564** | 0.01 | 0.0 | 0.563 | 0.4 | 0.0 |
| | 5 | **0.638** | 0.4 | 0.95 | 0.627 | 0.7 | 0.95 |
| | 6 | **0.600** | 0.3 | 0.0 | 0.590 | 0.7 | 0.0 |
| | 7 | **0.648** | 0.1 | 0.95 | 0.644 | 0.01 | 0.95 |
| | 8 | **0.642** | 0.3 | 0.0 | 0.624 | 1.2 | 0.0 |
| | 9 | 0.717 | 1.0 | 0.95 | **0.718** | 0.01 | 0.95 |
| | avg. | **0.639** | – | – | 0.632 | - | – |
| Small-Norb | 0 | 0.512 | 0.1 | 0.0 | **0.520** | 7.0 | 0.0 |
| | 1 | **0.656** | 0.01 | 0.0 | 0.647 | 0.01 | 0.0 |
| | 2 | **0.771** | 0.05 | 1.0 | **0.771** | 0.05 | 1.0 |
| | 3 | **0.581** | 10.0 | 1.0 | **0.581** | 10.0 | 1.0 |
| | 4 | **0.564** | 0.5 | 1.0 | **0.564** | 0.5 | 1.0 |
| | avg. | **0.617** | – | – | **0.617** | - | – |
| ECG (Arrhythmia) | 0 | **0.911** | 0.05 | 0.95 | 0.906 | 0.01 | 0.95 |
| | 1 | 0.924 | 0.01 | 0.95 | **0.925** | 0.01 | 0.95 |
| | 2 | **0.970** | 0.01 | 0.99 | **0.970** | 0.01 | 0.99 |
| | 3 | 0.905 | 0.01 | 0.95 | **0.910** | 0.01 | 0.95 |
| | 4 | **0.991** | 0.01 | 0.99 | **0.991** | 0.01 | 0.99 |
| | avg. | **0.940** | – | – | **0.940** | - | – |

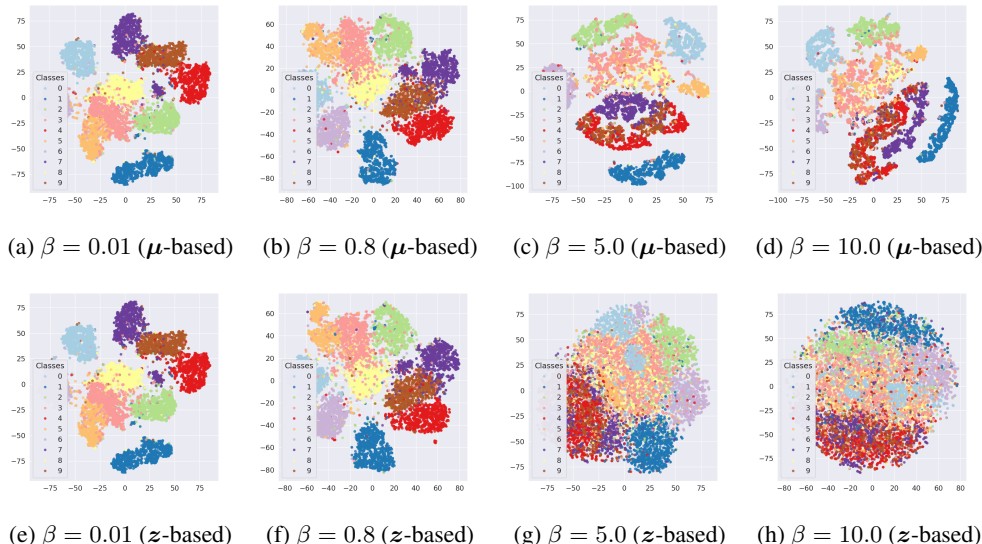

(a) $\beta = 0.01$ ($\mu$-based)    (b) $\beta = 0.8$ ($\mu$-based)    (c) $\beta = 5.0$ ($\mu$-based)    (d) $\beta = 10.0$ ($\mu$-based)

(e) $\beta = 0.01$ ($z$-based)    (f) $\beta = 0.8$ ($z$-based)    (g) $\beta = 5.0$ ($z$-based)    (h) $\beta = 10.0$ ($z$-based)

Figure 9: Comparison of $\mu$-based method or $z$-based method on MNIST data (anomalous class is 3) for inferring latent representations that are visualized in t-SNE map.

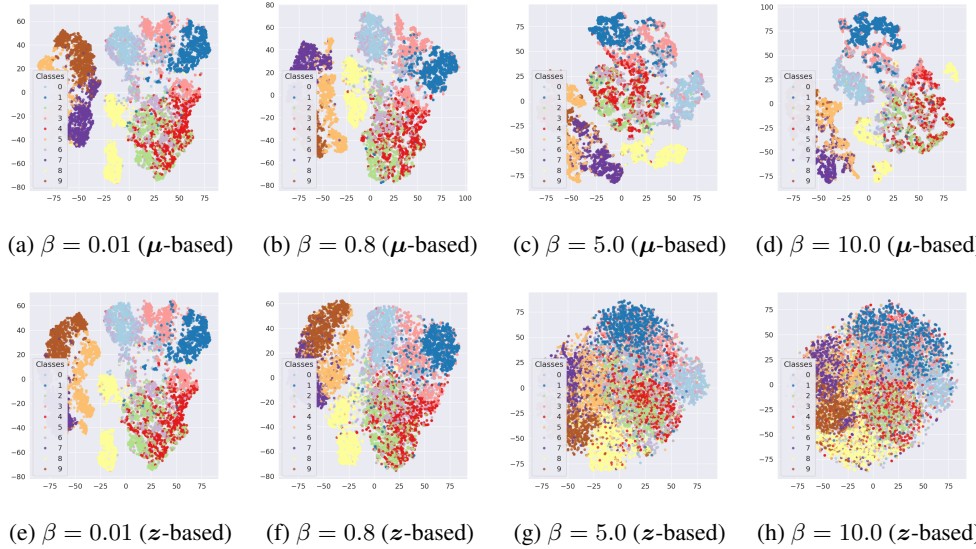

(a) $\beta = 0.01$ ($\mu$-based)    (b) $\beta = 0.8$ ($\mu$-based)    (c) $\beta = 5.0$ ($\mu$-based)    (d) $\beta = 10.0$ ($\mu$-based)

(e) $\beta = 0.01$ ($z$-based)    (f) $\beta = 0.8$ ($z$-based)    (g) $\beta = 5.0$ ($z$-based)    (h) $\beta = 10.0$ ($z$-based)

Figure 10: Comparison of $\mu$-based method or $z$-based method on Fashion-MNIST data (anomalous class is 1 ("Trouser/pants")) for inferring latent representations that are visualized in t-SNE space.

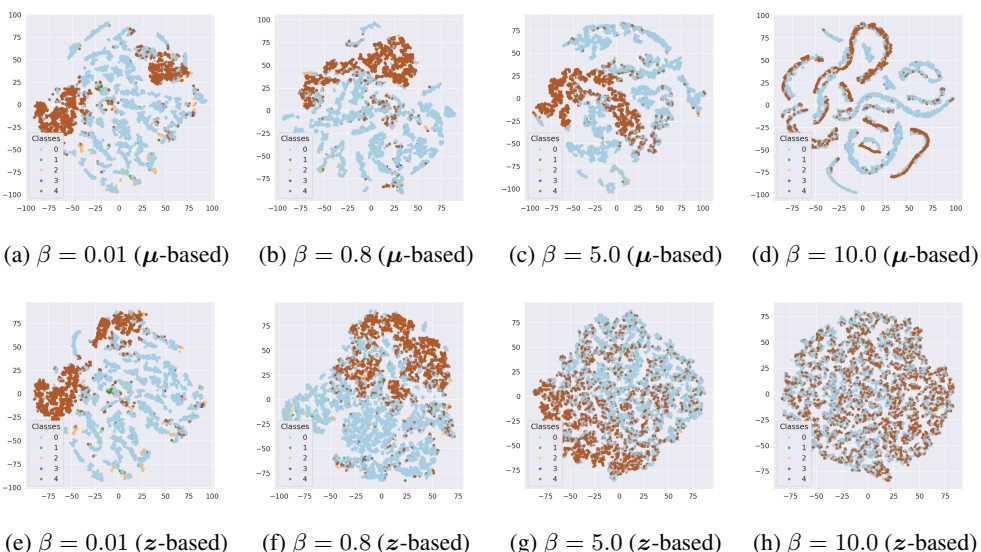

(a) $\beta = 0.01$ ($\boldsymbol{\mu}$-based)  (b) $\beta = 0.8$ ($\boldsymbol{\mu}$-based)  (c) $\beta = 5.0$ ($\boldsymbol{\mu}$-based)  (d) $\beta = 10.0$ ($\boldsymbol{\mu}$-based)

(e) $\beta = 0.01$ ($\boldsymbol{z}$-based)  (f) $\beta = 0.8$ ($\boldsymbol{z}$-based)  (g) $\beta = 5.0$ ($\boldsymbol{z}$-based)  (h) $\beta = 10.0$ ($\boldsymbol{z}$-based)

Figure 11: Comparison of $\boldsymbol{\mu}$-based method or $\boldsymbol{z}$-based method on Arrhythmia time-series data (anomalous class is 4 ("Q")) for inferring latent representations that are visualized in t-SNE space.

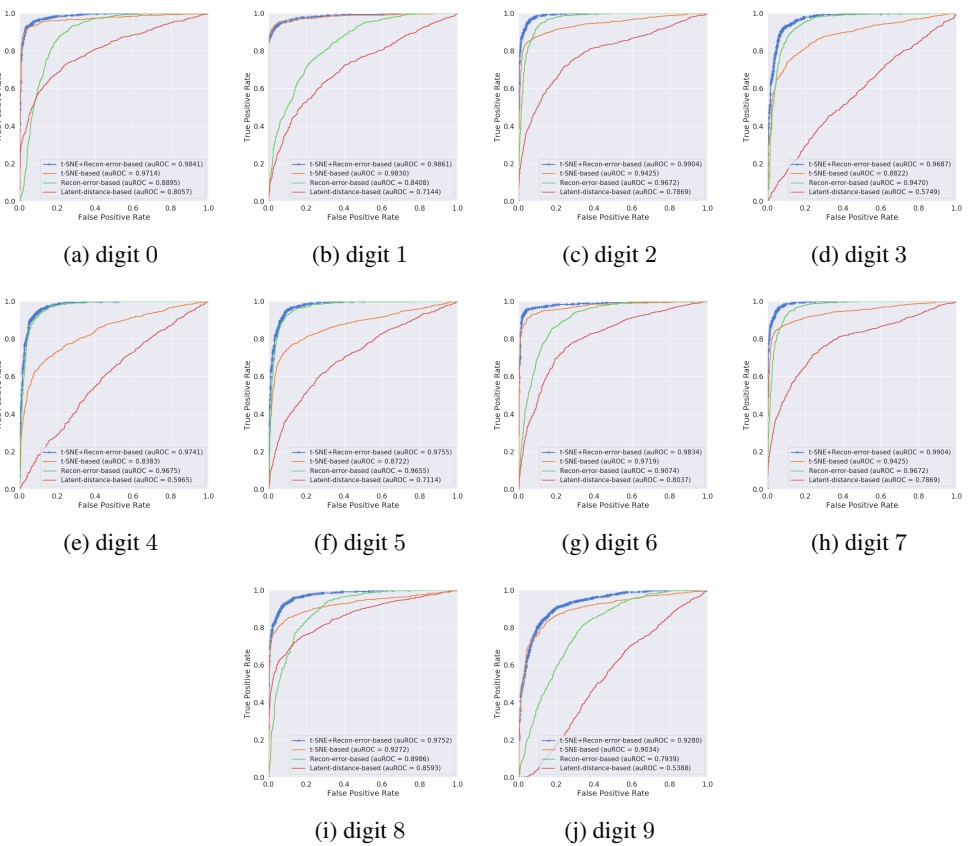

(a) digit 0  (b) digit 1  (c) digit 2  (d) digit 3

(e) digit 4  (f) digit 5  (g) digit 6  (h) digit 7

(i) digit 8  (j) digit 9

Figure 12: ROC plots on MNIST ($\boldsymbol{\mu}$-based).

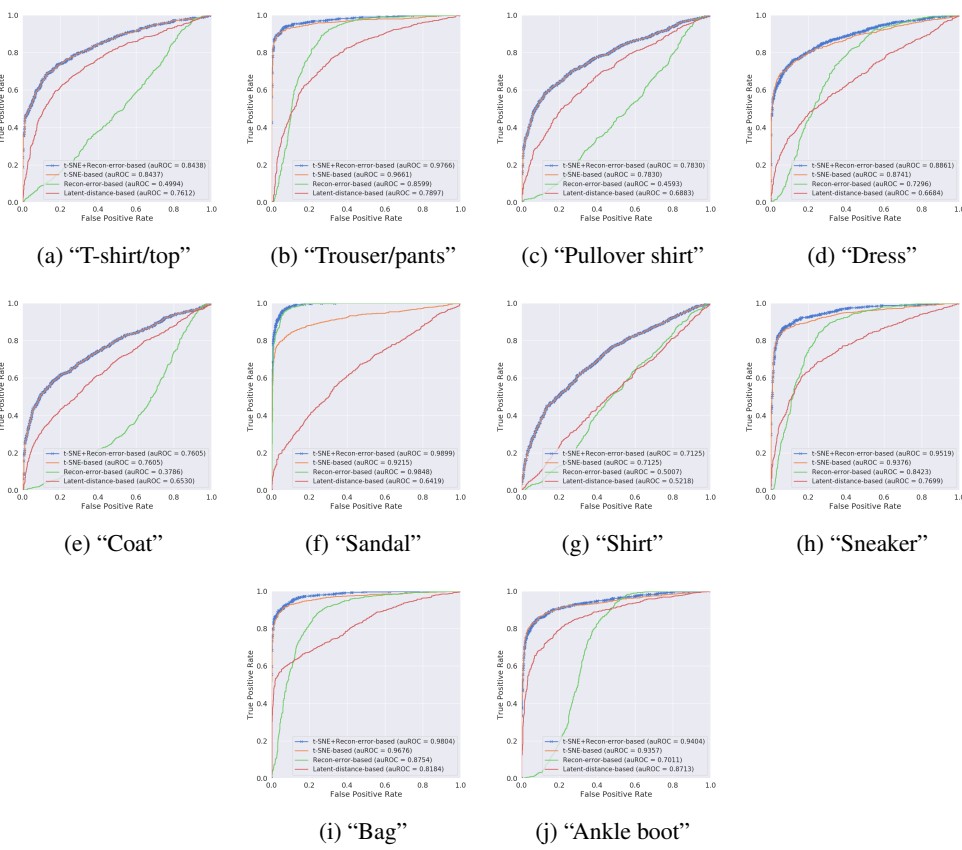

Figure 13: ROC plots on Fashion-MNIST ($\mu$-based).

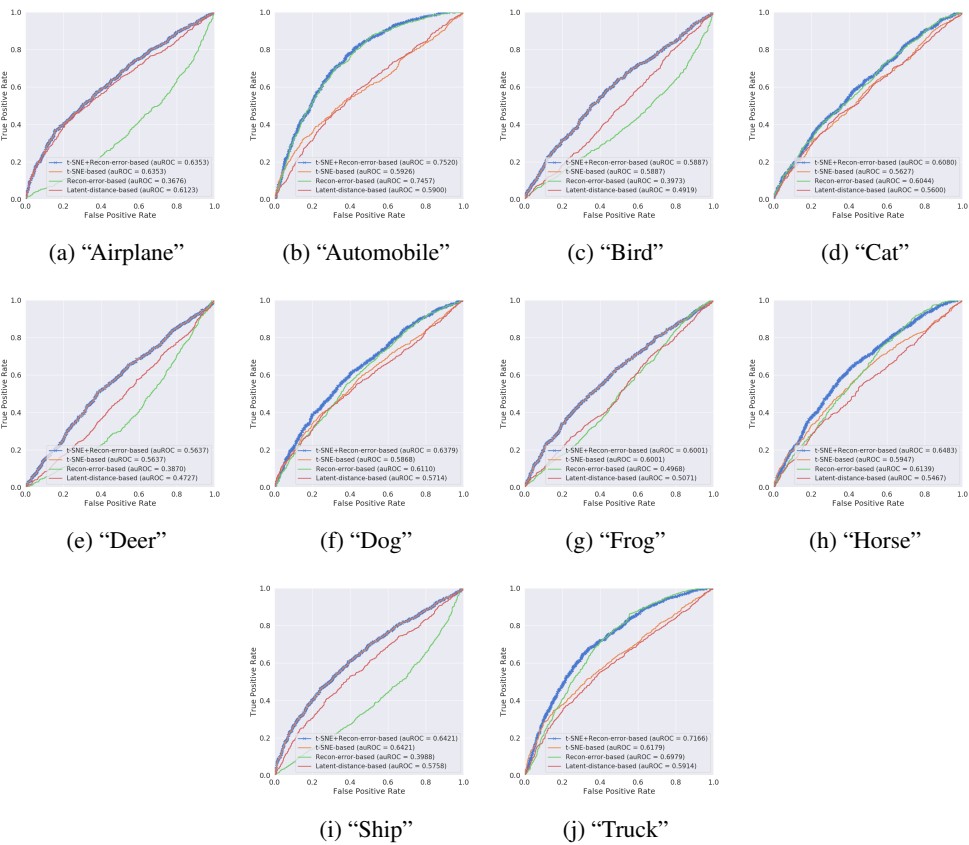

Figure 14: AuROC plots on CIFAR-10 ($\mu$-based).

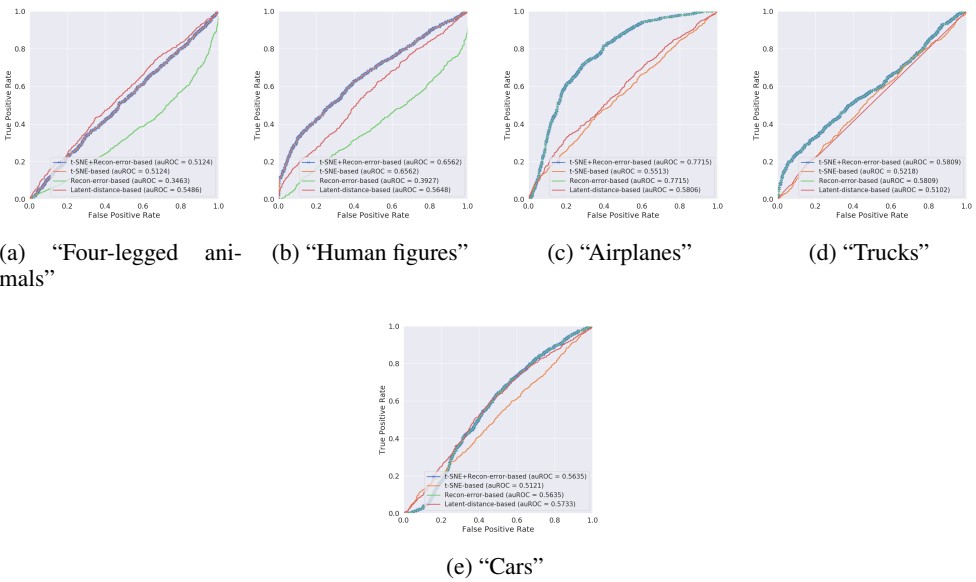

Figure 15: ROC plots on Small-Norb ($\mu$-based).

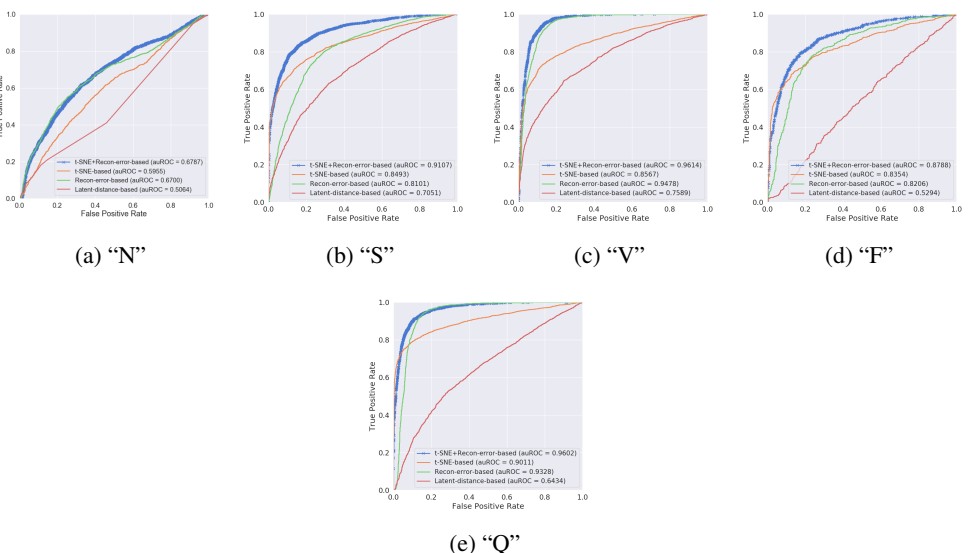

(a) "N"  (b) "S"  (c) "V"  (d) "F"

(e) "Q"

Figure 16: ROC plots of Arrhythmia using CNN-encoder ($\mu$-based).

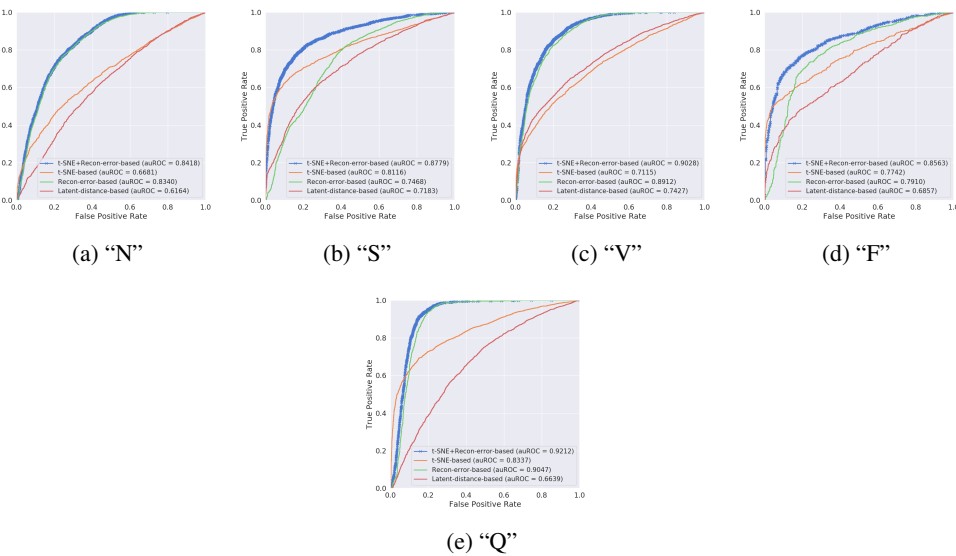

(a) "N"  (b) "S"  (c) "V"  (d) "F"

(e) "Q"

Figure 17: ROC plots of Arrhythmia using LSTM-encoder ($\mu$-based).

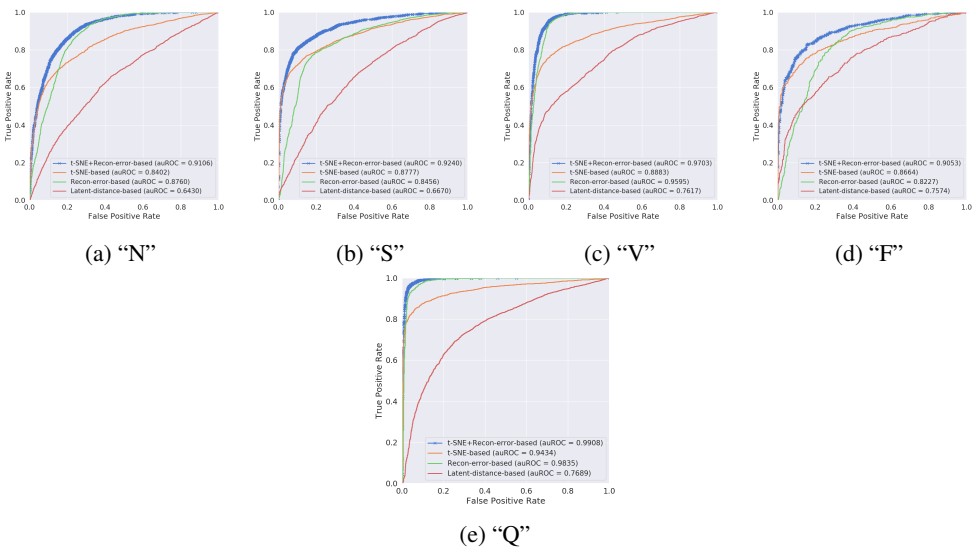

(a) "N"    (b) "S"    (c) "V"    (d) "F"

(e) "Q"

Figure 18: ROC plots of ECG using TCN-encoder ($\boldsymbol{\mu}$-based).

Table 3: Architecture of the $\beta$-VAE used in AnoDM.

| Dataset | Optimizer | | Architecture |
|---|---|---|---|
| **MNIST** | Adam ($1e^{-3}$) | Input | 784 (flattened $28 \times 28 \times 1$). |
| | | Encoder | Conv2d (filters 32, kernel_size 3, strides 2), Conv2d (filters 32, kernel_size 3, strides 2), FC (128), FC (128). ReLU activation. |
| | | Decoder | FC (128), FC (1568), Deconv2d (filters 32, kernel_size 3, strides 2), Deconv2d (filters 1, kernel_size 3, strides 2). ReLU activation. |
| **Fashion-MNIST** | Adam ($1e^{-3}$) | Input | 784 (flattened $28 \times 28 \times 1$). |
| | | Encoder | Conv2d (filters 32, kernel_size 3, strides 2), Conv2d (filters 32, kernel_size 3, strides 2), FC (128), FC (128). ReLU activation. |
| | | Decoder | FC (128), FC (1568), Deconv2d (filters 32, kernel_size 3, strides 2), Deconv2d (filters 1, kernel_size 3, strides 2). ReLU activation. |
| **CIFAR-10** | Adam ($1e^{-3}$) | Input | 3072 (flattened $32 \times 32 \times 3$). |
| | | Encoder | Conv2d (filters 64, kernel_size 3, strides 2), BN, Conv2d (filters 128, kernel_size 5, strides 2), BN, Conv2d (filters 128, kernel_size 5, strides 2), BN, FC (256), Dropout (0.9), BN, FC (256), Dropout (0.9), BN. ReLU activation. |
| | | Decoder | FC (256), BN, FC (256), BN, FC (2048), BN, Deconv2d (filters 128, kernel_size 5, strides 2), BN, Deconv2d (filters 64, kernel_size 5, strides 2), BN, Deconv2d (filters 3, kernel_size 3, strides 2). ReLU activation. |
| **Small-Norb** | Adam ($1e^{-3}$) | Input | 1024 (flattened $32 \times 32 \times 1$). |
| | | Encoder | Conv2d (filters 32, kernel_size 3, strides 2), BN, Conv2d (filters 64, kernel_size 3, strides 2), BN, Conv2d (filters 128, kernel_size 3, strides 2),BN, FC (256), Dropout (0.7), BN, FC (256), Dropout (0.7), BN. ReLU activation. |
| | | Decoder | FC (256), BN, FC (256), BN, FC (2048), BN, Deconv2d (filters 128, kernel_size 5, strides 2), Deconv2d (filters 64, kernel_size 5, strides 2), Deconv2d (filters 3, kernel_size 3, strides 2). ReLU activation. |
| **Arrhythmia** | Adam ($1e^{-3}$) | Input | 187. |
| | | Encoder | TCN (filters 64, kernel_size 4, stacks 1, dilations $[1, 2, 4, 8, 16, 32]$, padding "causal", use_skip_connections "True", dropout_rate 0.05) |
| | | Decoder | FC (256), FC (384), FC (2048), BN, Deconv1d (filters 64, kernel_size 3, strides 2), Deconv2d (filters 64, kernel_size 3, strides 2), Deconv2d (filters 64, kernel_size 3, strides 2), Deconv2d (filters 32, kernel_size 3, strides 2), Deconv2d (filters 32, kernel_size 3, strides 2), Deconv2d (filters 1, kernel_size 3, strides 2). ReLU activation. |

