# OpenReview forum: "Anomaly Detection Based on Unsupervised Disentangled Representation Learning in Combination with Manifold Learning"
_ICLR.cc/2020/Conference — Reject_

### Official Review · AnonReviewer2 · 2019-10-15
**Official Blind Review #2**

**Rating:** 3

**Review:**

In a paper a new way to compute anomality score (for a test point) is suggested. A paper is purely experimental, based on existing techniques to dimension reduction (beta-VAE and t-SNE). Given trained beta-VAE, latent vectors, obtained for training set, are feed into t-SNE algorithm. The overall anomality score for a test point is combined from 1-NN distances on t-SNE plot and reconstruction error of beta-VNE.

There is substantive question naturally appears from the application of t-SNE to obtained latent vectors. By construction, beta-VAE tries to make latent vector to be distributed according to N(0,I). By definition, it is very hard to project such a distribution on a plane, even by non-linear methods, such as t-SNE. Yet at the second step of paper's approach, this vectors are feed into 2-dimensional t-SNE.

This aspect makes me to think that optimal alpha in paper's anomality score should be close to 1. This would imply that t-SNE step is not needed at all. So, I am curious what actual optimal alpha was in experiments?

Or, how would results change if t-SNE mapping would be set to Identity tranformation (into space whose dimension is the same as latent space), but formula for anomality remains the same?

**Experience Assessment:**

I have published one or two papers in this area.

**Review Assessment: Checking Correctness Of Derivations And Theory:**

I assessed the sensibility of the derivations and theory.

**Review Assessment: Checking Correctness Of Experiments:**

I assessed the sensibility of the experiments.

**Review Assessment: Thoroughness In Paper Reading:**

I made a quick assessment of this paper.

---

> ### Author Response · Authors · 2019-11-05
> **Right extent of disentanglement is essential; ablation studies show t-SNE is critical**
>
> Thanks for the questions regarding disentanglement and t-SNE. We properly address them below. Hopefully it will convince you for a better rating.
>
> The right extent of disentanglement is essential in beta-VAE for anomaly detection. If the value of beta is not inappropriately large, the divergence between the inference distribution and prior N(0,I) is always greater than 0. That’s why we searched the value of beta, and interestingly found that best performance was surprisingly (because existing work focused on beta>1) obtained when beta<1, that is the push/approximation from the inference distribution towards N(0,I) is actually less than vanila VAE. A possible future work is to follow Mathieu et al. (2019) which redefined disentanglement as decomposition rather than independent through a structured prior. However, it may be practically challenging, but very interesting.
>
> As the discussion with Reviewer #3, in Figure 2, we can see the best results are reached before alpha=1. First, we clarify that, it does not mean that the reconstruction of beta-VAE play a dominant role and t-SNE useless; the main reason is that the normalised reconstruction error in input space and the distance in t-SNE map have very different magnitudes (as mentioned in Section 3.5). This can be evidenced from the scales of all t-SNE plots in the paper. Second, from Table 1, we can see that the performances of AnoDM without using t-SNE (alpha=1) are worse than that using t-SNE (alpha<1). Third, it is very important to mention that, from Table 2 (appendix), we can see that 37 out of 40 optimal results were obtained when alpha<1. The three cases where alpha=1 only occurred on the Small-Norb data where convolutional generative models could not learn good latent representations. In summary, t-SNE does play an essential role in AnoDM.  This is an interesting point to have identical number of dimensions in latent space and t-SNE map. Unfortunately, t-SNE can only map to 2 or 3 dimensions, and if we reduce the latent dimensionality to 2 or 3, it would certainly affect reconstruction quality in input space.

---

> > ### Author Response · Authors · 2019-11-15
> > **Regarding value of alpha**
> >
> > We have uploaded a revised paper with major clarifications and changes highlighted in red colour. Since large beta value would increase the overlap of clusters in the latent space, our experimental results show that optimal value of beta is small, in interval (0,1], which leads to good separations of clusters in latent space and stable model learning, while not strong enough to push the inference distribution too close to N(0,I).
> >
> > Regarding value of alpha, we added Appendix F in the new version. We  replaced the distance score in the combined anomaly score function with a distance score normalised by average distances among training samples in t-SNE map, and found that the optimal alpha value became much smaller. It implies that t-SNE is essentially needed in AnoDM. Please see Appendix F and Figure 8 for details.

---

### Official Review · AnonReviewer3 · 2019-10-15
**Official Blind Review #3**

**Rating:** 3

**Review:**

This paper proposes to combine beta-VAE and t-SNE for anomaly detection.
Although the problem and the proposed approach is relevant, I have the following concerns.

- The problem setting is not well explained.
    In particular, it is not clear whether the setting is unsupervised or not.
    It seems that the proposed method is for unsupervised anomaly detection.
    However, the authors mention that beta-VAE is trained on normal data, which means that it is not unsupervised.
    Please clarify this point.
- The originality and the technical quality of the proposed method is not high as it is a straightforward combination of two existing method.
    If the proposed combination has some theoretical advantage for anomaly detection, the paper becomes more interesting.
    However, there is no theoretical analysis of the proposed method, hence the significance of the contribution is hot high in its current state.
- Experimental results are not convincing.
    * The authors argue that the t-SNE step is important as the proposed method is better than the naive beta-VAE.
      However, Figure 2 shows that in most cases the score becomes better as \alpha gets larger and shows the best score if alpha = 1.
      From Equation (2), this means that the t-SNE step does not contribute to the anomaly detection performance.
      This inconsistency should be carefully discussed.
    * Also, Figure 2 shows that beta should be small and the best score is achieved when beta = 0 in most cases.
      This means that representation learning is not meaningful and the raw representation (feature vectors) may be already effective for anomaly detection.
      Hence the significance of the proposed method is not convincing.


**Experience Assessment:**

I have published one or two papers in this area.

**Review Assessment: Checking Correctness Of Derivations And Theory:**

I assessed the sensibility of the derivations and theory.

**Review Assessment: Checking Correctness Of Experiments:**

I carefully checked the experiments.

**Review Assessment: Thoroughness In Paper Reading:**

I read the paper at least twice and used my best judgement in assessing the paper.

---

> ### Author Response · Authors · 2019-11-05
> **AnoDM is unsupervised; the two components mutually complement; best results were obtained with alpha<1 and beta>0**
>
> Thank you for your time to review our paper and raise thoughtful questions for discussion. In the following, we address the three major concerns. We will add them in the revised paper. Better score please :)
>
> Our AnoDM framework is unsupervised. It means that beta-VAE is trained to model the distribution of the “normal” data points, and then used as a reference to identify out-of-distribution data points through reconstruction error and distance. The soul is similar to traditional statistical methods, but we apply deep models and representation learning for complex data. It is not supervised, because normal and anomalous samples are not treated equally as in two-class classification.  In the whole training process, we didn’t use any  labels of data, except when plotting the t-SNE graphs.
>
> Yes, it is relatively simple to combine two existing complementary methods together to obtain promising results for anomaly detection. Isn’t it the right need in real applications? Theoretically we highlight that the two methods complement each other: reconstruction error provides useful information in input space, while distance in t-SNE map contribution anomaly score from  latent space. Furthermore, this paper contributes a comprehensive empirical study about the impact of extent of disentanglement for anomaly detection. Although the AnoDM framework is technically straightforward, but it is practically valuable. It paves the road for more sophisticated studies and improvement on AnoDM. The two components could be replaced with new disentanglement learning and manifold learning algorithms. As future work, the two components can be combined in a single objective to learn a single model, rather than a two-phase framework.
>
> Thanks for opening this interesting discussion. In Figure 2, we can see the best results are reached before alpha=1. First, we clarify that, it does not mean that the reconstruction of beta-VAE play a dominant role and t-SNE useless; the main reason is that the normalised reconstruction error in input space and the distance in t-SNE map have very different magnitudes (as mentioned in Section 3.5). This can be evidenced from the scales of all t-SNE plots in the paper. Second, from Table 1, we can see that the performances of AnoDM without using t-SNE (alpha=1) are worse than that using t-SNE (alpha<1). Third, it is very important to mention that, from Table 2 (appendix), we can see that 37 out of 40 optimal results were obtained when alpha<1. The three cases where alpha=1 only occurred on the Small-Norb data where convolutional generative models could not learn good latent representations (CapsNet does). In summary, t-SNE does play an essential role in AnoDM.  It is also important to clarify that best results were achieved with 0<beta<1. We will improve the visualisation in Figure 2.

---

> > ### Comment · AnonReviewer3 · 2019-11-14
> > **Post-Reviews Update**
> >
> > Thank you for clarifying my concerns.
> > - I understand that the problem setting is unsupervised.
> > - A simple combination is fine if its effectiveness is convincing and carefully evaluated, while currently it is not the case due to the lack of theoretical analysis.
> > - I also understand that \alpha = 1 is not the optimal. I strongly recommend to improve visualization in Figure 2. I think 2D plots are easier to see.
> > - I see that \beta = 0 is not the optimal, while it is still true that quite small \beta values give the optimal results. More careful evaluation is required.
> > I increase my score to weak reject, while I still think the paper is sill below the acceptance threshold.

---

> > > ### Author Response · Authors · 2019-11-15
> > > **Heatmaps in Figure 2; Analysis for small beta values**
> > >
> > > Thank you for considering our feedback. Now we just uploaded a revision where major changes are highlighted by red colour. We now use 2D heatmaps in Figure 2 (thanks for this suggestion!) which better show that small beta values in (0,1] give optimal results (note the upper right area of a heatmap). It can be explained that, according to (Mathieu et al. 2019), larger beta value increases the overlap of clusters in latent space. Thus small beta values would encourage separation of clusters, but beta=0 could make model learning highly unstable because it loses control to the variance in latent variables.
> > >
> > > We also replaced the distance score in the combined anomaly score function with a distance score normalised by average distances among training samples in t-SNE map, and found that the optimal alpha value became much smaller. It implies that t-SNE is essential in AnoDM. Please see Appendix F and Figure 8 for details.
> > >
> > > Hope the revised paper could convince you further.

---

### Official Review · AnonReviewer1 · 2019-10-21
**Official Blind Review #1**

**Rating:** 6

**Review:**

This paper presents a novel deep anomaly detection model. It combines two existing models: B-VAE and t-SNE. The B-VAE is trained unsupervised and learns an encoder and decoder which provide both an embedding and a reconstruction. Using t-SNE to reduce its dimensionality, the embedding is projected into a 2 dimensional space. An anomaly score function is defined that combines the reconstruction error and the distance in t-SNE space to the K nearest neighbor(s). Experiments are conducted with several image datasets (MNIST,FMNIST,CIFAR10,SmallNORB) and one timeseries dataset (Arrhythmia). For the image sets, the B-VAE model is implemented with a CNN, while for timeseries, a TCN is used. Comparisons are conducted showing the approach to be beat other SOT unsupervised methods, AnoGAN and ADGAN, by 63% and 22% respectively for MNIST and 8% and 2% for FMNIST (in terms of error reduction). For CIFAR-10 and FMNIST it is even demonstrated to beat a supervised SOT method CapsNET. Another experiment shows that t_SNE dramatically improves the performance over B-VAE alone. For the timeseries, the approach is not compared to other SOT approaches as the authors only provide an experiment showing that TCN beats CNN and LSTM for the implementation of the B-VAE. In addition the authors study the effect of the various parameters of the system, in particular the effect of the B in B-VAE and of alpha, the mixing factor between reconstruction error and kNN distance in t_SNE. 3D plots give a good idea on how to select optimal values for the various datasets. The impact of B is also shown on the t-SNE map for MNIST. Finally an ablation studies compares on MNIST the performance of the approach with t-SNE alone, reconstruction alone, and latent distance. On average over 4 digits taken as anomaly, the proposed approach dramatically outperforms the others.

PROS:

* The proposed approach improves over the SOT of competitive recent methods for anomaly detection on four image datasets.
* The authors make an effort to abstract the approach into a framework where other deep learning models and dimensionality reduction techniques can be used. They illustrate this by using a TCN instead of a CNN for the timeseries example.
* The parameter studies and ablation studies are informative and answer many of the questions i had as i read the paper.
* The paper is relatively clearly written (at least sufficiently to easily understand the technical details).


CONS:

* The novelty of the paper is limited as it is mostly a combination of 2 existing methods.
* The timeseries dataset is not compared to SOT methods (although the authors claim SOT in the conclusion).
* A pseudo-code algorithm is not provided, making it unlikely someone can reproduce the method.
* There are many typos an grammatical errors
* The paper could have been shortened. 10 pages is too long.

Overall, because of the good performance and thoughtful ablation studies, and despite the limited novelty, I think the paper makes a good contribution to anomaly detection.



**Experience Assessment:**

I have published one or two papers in this area.

**Review Assessment: Checking Correctness Of Derivations And Theory:**

I assessed the sensibility of the derivations and theory.

**Review Assessment: Checking Correctness Of Experiments:**

I carefully checked the experiments.

**Review Assessment: Thoroughness In Paper Reading:**

I read the paper thoroughly.

---

> ### Author Response · Authors · 2019-11-05
> **Thanks for your support**
>
> Thanks for your support. We will do the following to improve this work. (1) We will clarify that since existing work uses CNN/LSTM-VAEs (e.g. Park et al. 2017) which are special cases of beta-VAE, thus as long as best performance is achieved when beta is not 1 and alpha is not 1, it means the AnoDM  framework outperforms SOT methods on the time-series data. (2) A pseudo-code algorithm is actually provided in appendix. (3) We will carefully go through the paper to correct typos and grammatical errors. (4) We actually shorten the paper from 16 to 10 pages before submitting the first version. It is difficult to shrink it again because lots of key information have to stay. We will try again though.

---

> > ### Author Response · Authors · 2019-11-15
> > **Revision submitted**
> >
> > We have uploaded a better version to address all reviews' comments.

---

### Decision · Program_Chairs · 2019-12-19

**Decision:**

Reject

**Comment:**

The paper presents AnoDM (Anomaly detection based on unsupervised Disentangled representation learning and Manifold learning) that combine beta-VAE and t-SNE for anomaly detection. Experiment results on both image and time series data are shown to demonstrate the effectiveness of the proposed solution.

The paper aims to attack a challenging problem. The proposed solution is reasonable. The authors did a job at addressing some of the concerns raised in the reviews. However, two major concerns remain: (1) the novelty in the proposed model (a combination of two existing models) is not clear; (2) the experiment results are not fully convincing. While theoretical analysis is not a must for all models, it would be useful to conduct thorough experiments to fully understand how the model works, which is missing in the current version.

Given the two reasons above, the paper did not attract enough enthusiasm from the reviewers during the discussion. We hope the reviews can help improve the paper for a better publication in the future.